# Spatial dynamics of CD39$^+$CD8$^+$ exhausted T cell reveal tertiary lymphoid structures-mediated response to PD-1 blockade in esophageal cancer

Kenro Tanoue [1], Hirofumi Ohmura [1,2], Koki Uehara[1], Mamoru Ito[1], Kyoko Yamaguchi [1], Kenji Tsuchihashi[1], Yudai Shinohara[3], Peng Lu [4], Shingo Tamura[5], Hozumi Shimokawa[6], Taichi Isobe [1,2], Hiroshi Ariyama[7], Yoshihiro Shibata[8], Risa Tanaka[9], Hitoshi Kusaba[6], Taito Esaki[10], Kenji Mitsugi[11], Daisuke Kiyozawa[12], Takeshi Iwasaki [12], Hidetaka Yamamoto[12,13], Yoshinao Oda[12], Koichi Akashi[1] & Eishi Baba [2] ✉

Despite the success of immune checkpoint blockade (ICB) therapy for esophageal squamous cell cancer, the key immune cell populations that affect ICB efficacy remain unclear. Here, imaging mass cytometry of tumor tissues from ICB-treated patients identifies a distinct cell population of CD39$^+$PD-1$^+$CD8$^+$ T cells, specifically the TCF1$^+$ subset, precursor exhausted T (CD39$^+$ Tpex) cells, which positively correlate with ICB benefit. CD39$^+$ Tpex cells are predominantly in the stroma, while differentiated CD39$^+$ exhausted T cells are abundantly and proximally within the parenchyma. Notably, CD39$^+$ Tpex cells are concentrated within and around tertiary lymphoid structure (TLS). Accordingly, tumors harboring TLSs have more of these cells in tumor areas than tumors lacking TLSs, suggesting Tpex cell recruitment from TLSs to tumors. In addition, circulating CD39$^+$ Tpex cells are also increased in responders following ICB therapy. Our findings show that this unique subpopulation of CD39$^+$PD-1$^+$CD8$^+$ T cells is crucial for ICB benefit, and suggest a key role in TLS-mediated immune responses against tumors.

Immunotherapy, particularly the use of immune checkpoint blockade (ICB) drugs to target PD-1 or CTLA4, has significantly improved the prognosis for esophageal cancer[1–5]. PD-1 inhibitors are not only used as monotherapy; they can also be also combined with CTLA4 inhibitors and cytotoxic agents, establishing them as key components of systemic chemotherapy for esophageal cancer[1,5]. However, not all patients benefit from ICB therapy. Specifically, PD-L1 expression has proven less effective for use as a reliable biomarker for predicting survival benefits from ICB therapy[1,3,4]. Given the crucial role of CD8$^+$ T cells as mediators of ICB, a comprehensive analysis to identify the determinants of therapeutic success is essential.

While tumor-reactive CD8$^+$ T cells are potential targets of ICB in human cancers, the identification of tumor antigen-specific T cells in primary samples has remained challenging, limiting further elucidation of immune mechanisms. CD39 has recently emerged as a promising surrogate marker for these calls[6–11], playing an integral role in the immune response against various solid tumors[12,13]. These cells are characterized by oligo-clonality, an exhausted phenotype[6,14–18], and a highly proliferative state[6,8,12,13,17], and can be systemically and locally reinvigorated by ICB therapy[19,20].

Exhausted T (Tex) cells exhibit significant heterogeneity, including 'stem-like' exhausted precursor (Tpex) cells that express T cell

factor 1 (TCF-1) and differentiate into a late, dysfunctional cell state[21–26]. Due to their high capacity for proliferation and self-renewal, Tpex cells are associated with clinical benefits in ICB therapy[24,25] and are expected to localize in specific areas within tumor tissues. However, the spatial characteristics of Tpex cells that infiltrate tumor tissues, as potential targets of ICB, remain unclear.

The proximity of certain immune cells to tumors or PD-L1+ antigen-presenting cells has clinical implications[27,28], suggesting that spatial analysis could provide deeper insights into intratumoral immune cell dynamics. Moreover, TCF-1+ T cells or PD-1hiCD8+ T cells are predominantly distributed within tertiary lymphoid structures (TLSs)[29–31]. Given the complexity of the tumor microenvironment (TME), which includes TLSs, elucidating the spatial distribution of CD39+CD8+ T cells within the TME, particularly regarding their dysfunctional states, is crucial for a better understanding of the underlying immune landscape of tumor-reactive CD8+ T cells.

In this work, we reveal a dynamic immune landscape in esophageal squamous cell cancer (ESCC) mediated by TLSs through spatial proteomics using imaging mass cytometry (IMC), highlighted by the distinct distribution patterns of TCF-1+ subsets within intratumoral CD39+CD8+ Tex cells, which are closely associated with clinical benefits from ICB therapy. Further investigation into post-ICB immune kinetics in the blood suggests that the influence of these cells extends beyond TLSs, contributing to systemic anti-tumor immunity. Our study advances the understanding of the underlying immune landscape of tumor-reactive CD8+ T cells and has the potential to contribute to therapeutic strategy development in the field of precision oncology.

## Results

### PD-L1 expression did not correlate with response to ICB therapy

Among the 31 patients initially enrolled to receive nivolumab monotherapy, 30 patients with unresectable advanced or recurrent ESCC, who were refractory or intolerant to platinum-containing chemotherapy, were eligible for the tissue and peripheral blood analysis in this study (Fig. 1a, b). Of all 30 eligible patients, 7 (23.3%) achieved partial response (PR), and 10 (33.3%) patients (PR + long stable disease [SD]) were classified as clinical responders (Fig. 1c). No differences in combined positive score (CPS) were observed based on clinical response to ICB therapy (Fig. 1d). Consequently, it is considered crucial to characterize host immune cells rather than PD-L1 expression alone to identify factors associated with improvement in the efficacy of immunotherapies for ESCC.

### Tumor CD39+PD-1+ Tex cells are associated with the clinical benefit of ICB therapy

To further investigate the major determinants of ICB responses, we utilized multiplexed IMC to analyze the TME of 27 formalin-fixed paraffin-embedded (FFPE) pre-treatment tumor tissue samples (Fig. 2a, b). Next, we manually phenotyped the immune or epithelial cells within the tumor regions of interest (ROIs), determining each phenotype's proportion in 27 tumor-focused ROIs per patient (Fig. 2c, d). A significant link was noted between a high ratio of PD-1+CD8+ T cells to the total cell population and ICB therapy response ($P = 0.0045$; Fig. 2e), which is similar to previous reports[32], despite no significant difference in PD-1 expression rates on CD8+ T cells between responders and non-responders ($P = 0.1511$; Fig. S1C). UMAP plots of representative CD8+ T cells revealed a distinct subset of PD-1hi cells co-expressing TOX, Ki67, and CD39 (Fig. 2f), which are associated with T-cell exhaustion[14,16], proliferative activity, and tumor specificity[6–11], respectively. Additionally, the positivity of CD39 was higher in PD-1+CD8+ T cells in the TME than in peripheral blood mononuclear cells (PBMCs) ($P < 0.0001$; Fig. 2g). This result suggests that PD-1+CD8+ T cells, potentially reactive to tumors, are more prevalent in the TME and that PD-1 blockade could therefore be beneficial.

We next identified CD39+PD-1+ Tex cells (CD39+PD-1+CD8+) (Fig. 3a). These cells were characterized by high Ki67 and TOX expression ($P < 0.0001$; Fig. 2h), outperforming bystander T cells (CD39−PD-1−CD8+). The density of CD39+PD-1+ Tex cells was predictive of the ICB response ($P = 0.0488$; Fig. S1D), and CD39 expression in PD-1+CD8+ T cells tended to be higher in responders ($P = 0.0767$; Fig. S1E). This proliferating and exhausted CD39+PD-1+CD8+ population likely corresponds to tumor-reactive T cells, thereby contributing to the anti-tumor ICB response[6–8,13].

### CD39+ Tpex cells identified by TCF-1 expression constitute a distinct group among CD39+PD-1+ Tex cells

We hypothesized that the phenotypes of CD8+ T cells might differ within the cohort of CD39+PD-1+ Tex cells. From 11 independent ROIs, each containing more than 500 CD8+ T cells, we applied unsupervised clustering to CD8+ T cells using FlowSOM (Fig. S2A) and visualized these clusters in UMAP plots (Figs. S2B, S2C). We identified five CD39hiPD-1hi Tex clusters (C1–5) characterized by high TOX and PD-1 expression (Fig. S2A). When mapping these clusters to the representative ROIs of tumors lacking TLSs (Fig. S2D) and tumors containing TLSs (Fig. S2E), clusters C1 and C2, which exhibited high levels of TCF-1 expression and correspond to Tpex cells due to their exhaustion profile, were primarily localized to the stroma and TLSs. The localization characteristics of these TCF-1+ populations to TLSs were consistent with previously reported findings[29,30]. In contrast, clusters C3–5, primarily localized within the tumor parenchyma, corresponded to differentiated Tex (dTex) cells lacking TCF-1 expression (Fig. S2D, S2E). The presence of CD39+ Tpex (TCF-1+CD39+PD-1+CD8+ T) cells, previously reported in several studies[26,33–35], was also confirmed through co-staining in the tumor ROIs (Figs. 3a, S11B). To characterize this unique population within the TME, we categorized CD39+PD-1+ Tex cells from 27 cases as CD39+ Tpex and CD39+ dTex (TCF-1−CD39+PD-1+CD8+ T) cells. Both subsets exhibited high exhaustion marker levels (Fig. S1A, S1B), suggesting that they are potentially tumor-reactive T cells under continuous antigen stimulation in the TME[14–17].

To obtain further substantiation of their existence, we analyzed a dataset[33] from a cohort of head and neck squamous cell carcinoma (HNSCC) patients treated with atezolizumab (anti-PD-L1), where tumor-infiltrating lymphocytes (TILs) were examined using mass flow cytometry (MC) (Fig. S3A). We identified populations of CD39hiPD-1hi clusters (Clusters; C1, C2, C3, C4) through unsupervised clustering of CD8+ T cells with FlowSOM (Fig. S3B) and visualized them using UMAP plots (Fig. S3C). Notably, clusters C3 and C4, which correspond to CD39+ Tpex cells, were characterized not only by their expression of TCF-1, but also their relatively higher expression of CCR7, CD127, and TIGIT (Figs. S3B, S3D), indicating that they are phenotypically distinct Tpex cells, unlike CD39+ dTex cells[24,26,31,33]. These results collectively indicate that CD39+PD-1+CD8+ T cells constitute a heterogeneous group that includes not only dTex cells but also Tpex cells.

### Abundance of CD39+ Tpex cells in tumors is associated with clinical response to ICB therapy

Previous studies have highlighted the correlation between intratumoral TCF-1+CD8+ T cells and clinical benefit from ICB therapy in melanoma[21,36] and breast cancer[37]. Given these findings, we explored this correlation in ESCC by analyzing the association between clinical outcomes and the abundance of CD39+ Tpex cells. The relative abundance of these cells among CD39+PD-1+ Tex cells in the tumor and stroma area (excluding TLS areas) showed no clear impact on clinical outcomes (Fig. S1F), which is consistent with previous melanoma findings[21]. However, a significant positive correlation was observed between the densities of CD39+ Tpex cells and CD39+PD-1+ Tex cells

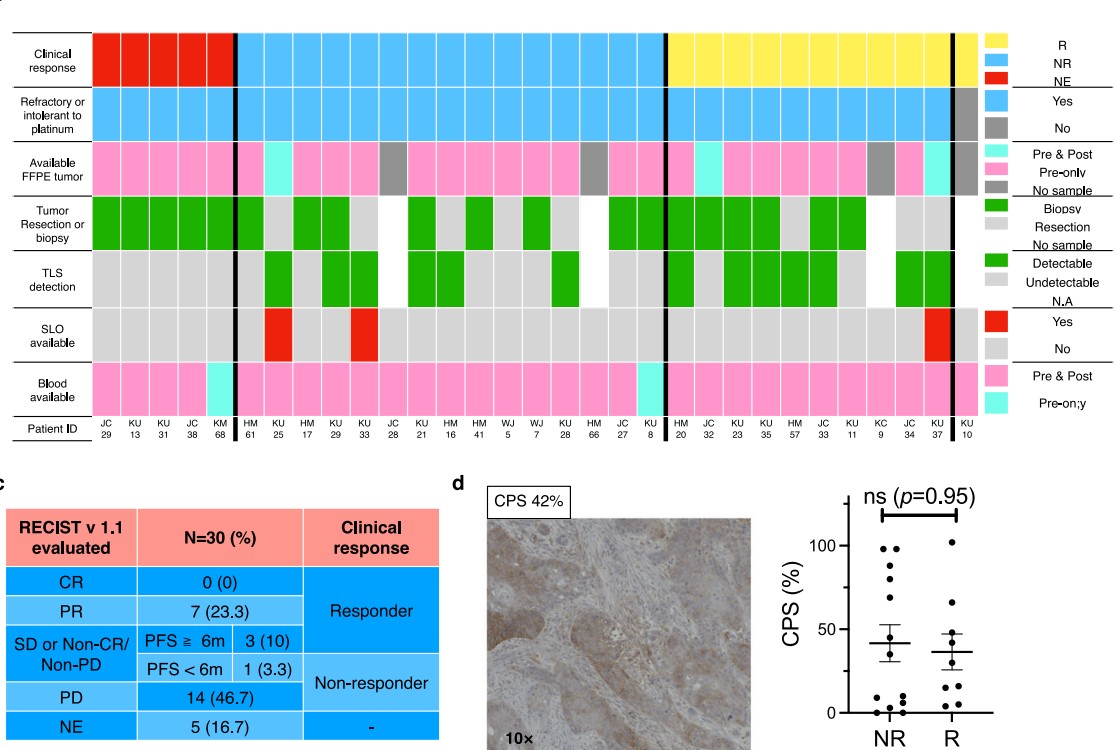

**Fig. 1 | Study design and clinical data for the 31 enrolled patients. a** Tumor, blood, and secondary lymphoid organ (SLO) collection from esophageal squamous cell carcinoma (ESCC) patients treated with nivolumab monotherapy for imaging mass cytometry (IMC), mass flow cytometry (MC), and multicolor flow cytometry (FC). Among the 31 enrolled patients, one patient (KU10) was ineligible for analysis due to no history of fluoropyrimidine-based and platinum-based chemotherapy. For more details, see the "Methods". **b** Clinical data and samples analyzed for each of the 31 ESCC patients. TLS: tertiary lymphoid structure, R: responder, NR: non-responder, NE: not evaluable, NA: not applicable. **c** Clinical outcomes of 30 eligible ESCC patients: responders (complete response; CR + partial response; PR + stable disease; SD or non-CR/non-PD ≥ 6 months), non-responders (SD or non-CR/non-PD < 6 months + progressive disease; PD). **d** Representative PD-L1 immunostaining for an ESCC patient. Combined positive score (CPS) in Rs (*n* = 9 patients) vs. NRs (*n* = 13 patients); two-sided Mann-Whitney *U*-test (*P* = 0.95). Five NE cases are excluded. PD-L1 staining across 27 independent samples was validated by a pathologist to ensure accuracy. **d** Error bars indicate mean ± SEM. **a** was created in BioRender. Kenro, T. (2024) BioRender.com/s50j163. Source data are provided as a Source Data file.

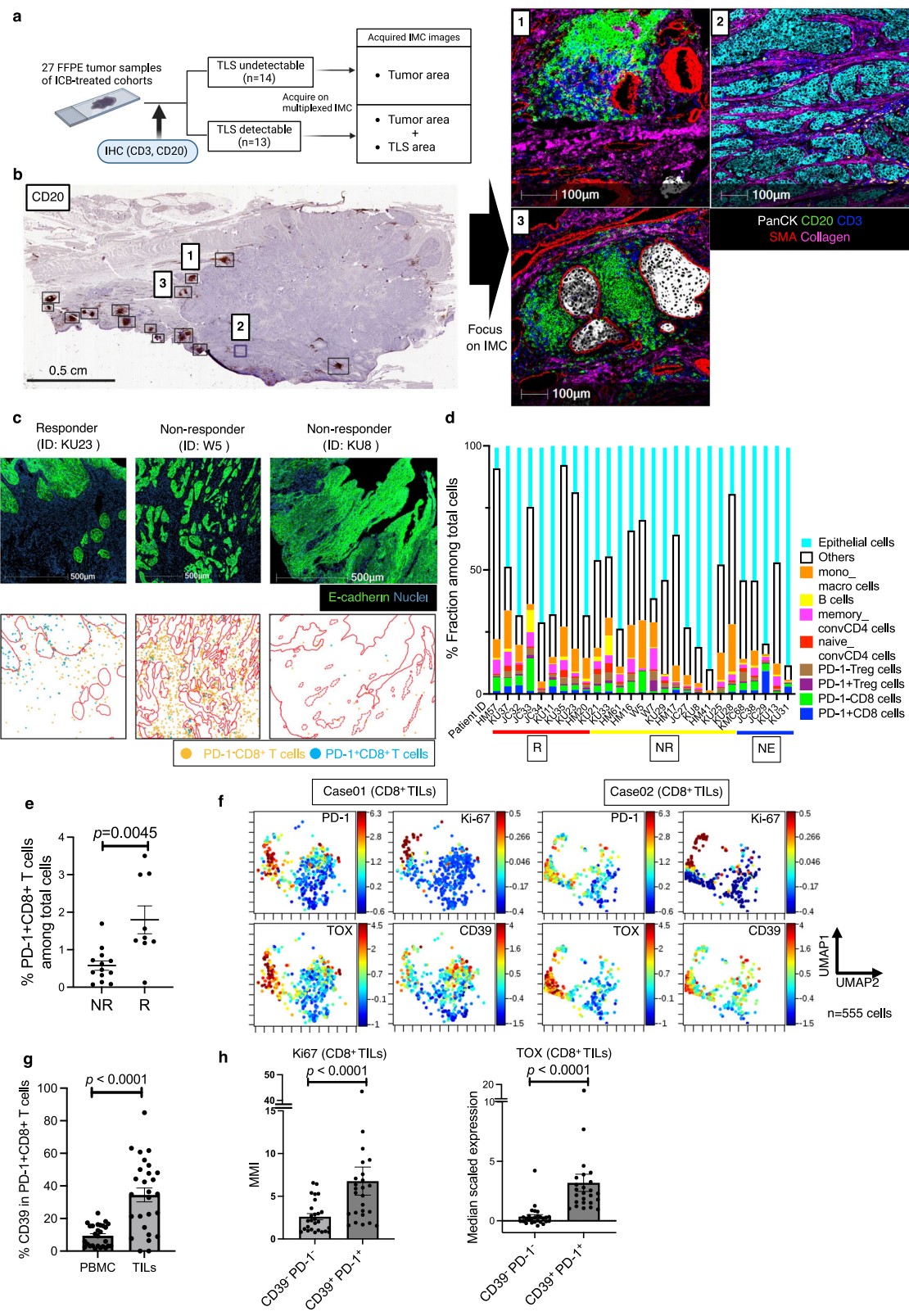

predictive of ICB therapy benefit ($n = 27$ patients, $r = 0.795$, $P < 0.001$; Fig. 3b). A similar correlation was also observed using the ICB cohort dataset[33] ($n = 8$ patients, $r = 0.831$, $P = 0.0106$; Fig. S3E). Accordingly, the absolute density of CD39[+] Tpex cells in the tumor and stroma area was significantly higher in ICB therapy responders than non-responders ($n = 22$ patients, $P = 0.0082$; Fig. 3c). These results suggest that CD39[+] Tpex cells, as potential precursors of CD39[+] dTex cells, constitute a significant component of the immune responses against the tumor.

## Tumor CD39[+]PD-1[+] Tex cells exhibit a distinct spatial organization that reflects their hierarchical differences

To explore the interplay between TME organization and tumor-reactive CD8[+] T cells, we investigated the spatial characteristics of

**Fig. 2 | CD39⁺PD-1⁺CD8⁺ T cells exhibit a proliferative exhausted phenotype.**
**a** Schematic illustration of IMC processing of tumor samples with or without tertiary lymphoid structures (TLSs). **b** IMC processing example for a tumor with TLSs (patient ID: KU37). The IMC images are pseudo-colored as follows: Pan-cytokeratin (PanCK) (white), CD20 (green), CD3 (blue), SMA (red), and Collagen (pink). **c** Spatial distribution images of PD-1⁻CD8⁺ (orange plots) and PD-1⁺CD8⁺ (blue plots) T cells, derived from three representative IMC images, pseudo-colored as follows: E-cadherin (green) and Nuclei (dark blue). Tumor parenchyma in distribution images is outlined in red. **d** Each fraction (%) of the annotated immune profile among total cells from the tumor-ROIs for 27 patients at pre-treatment. **e** Proportion (%) of PD-1⁺CD8⁺ T cells among total cells in tumor ROIs, compared between responders (R, *n* = 9) vs. non-responders (NR, *n* = 13); two-sided Mann-Whitney *U*-test (*P* = 0.0045). Five NE cases are excluded. **f** Two representative

UMAP plots of CD8⁺ tumor-infiltrating lymphocytes (TILs) (*n* = 555 cells each) within the tumor-ROIs, colored by the expression of each marker. The selected UMAP plots were chosen for their clear depiction of exhaustion patterns, which were observed in a subset of cases among a total of 11 independent ROIs.
**g** Proportion (%) of cells expressing CD39 among PD-1⁺CD8⁺ T cells in paired pretreatment PBMCs vs. TILs; two-sided Wilcoxon matched-pairs signed-rank test (*n* = 27, *P* < 0.0001). **h** Median mass intensity (MMI) of Ki67 and scaled expression of TOX in CD39⁺PD-1⁺CD8⁺ TILs vs. CD39⁻PD-1⁻CD8⁺ TILs; two-sided Wilcoxon matched-pairs signed test (each *n* = 25 independent tumor-ROIs, *P* < 0.0001 for both comparisons). Two cases with cold tumors lacking CD39⁺PD-1⁺CD8⁺ T cells are excluded. **e, g, h** Error bars indicate mean ± SEM. **e, g** *n* refers to patients. **a** was created in BioRender. Kenro, T. (2024) BioRender.com/s50j163. Source data are provided as a Source Data file.

CD39⁺PD-1⁺ Tex cells within the tumor area using samples from 27 enrolled patients. In this cohort, most T cells, regardless of type, were localized in abundance in the stroma over the parenchyma (Fig. 3e); however, approximately half (mean, 54.6%) of the CD39⁺ dTex cells were localized in the parenchyma, a rate higher than that of CD39⁺ Tpex cells (*P* = 0.0097; Fig. 3e). CD39⁺ Tpex cells were predominantly localized in the stroma (Fig. 3d, *P* = 0.0067; Fig. 3f), compared to the parenchyma, with a higher percentage of TCF-1 positivity within the stroma (*P* = 0.0237; Fig. 3g). Among cells within 400 *µm* of the parenchyma, a large subset was located even closer, within 40 *µm* (Fig. 4a). In results obtained from 27 individuals, CD39⁺ dTex cells were significantly more concentrated within the 0–10 *µm* range than the 30–40 *µm* range (**P* < 0.05; Fig. 4b). However, in this cohort, stromal CD39⁺ Tpex cells were localized relatively distant from the tumor compared with stromal CD39⁺ dTex cells (*P* = 0.0304; Fig. 4c, d), suggesting that exhaustion hierarchy impacts the cells' spatial characteristics. These data suggest that CD39⁺ dTex cells, which are more cytotoxic than Tpex cells and located closer to the tumor (Fig. S6C)²¹,²⁶,³³, interact with the tumor more directly. Considering that Tpex cells differentiate into dTex cells within the tumor³⁸, this suggests that stromal CD39⁺ Tpex cells migrate to the parenchyma as they differentiate into CD39⁺ dTex cells.

**Presence of TLSs comprising abundant CD39⁺ Tpex cells is associated with the hierarchical patterns of CD39⁺PD-1⁺ Tex cells within the TME**
Immunohistochemical (IHC) staining of CD3 and CD20 in FFPE tumor sections revealed detectable TLSs in 13 cases (Fig. 2a) associated with a high clinical response rate to ICB therapy (*n* = 27 patients, **P* < 0.05; Fig. 5a), a finding consistent with other cancers²⁹,³⁹. To explore the relationship with ICB therapy, we analyzed the immune composition within the TLS compared to the tumor area (Fig. 5b, c). Our findings revealed a higher density of naïve CD8⁺ T cells and PD-1⁺CD8⁺ T cells in the TLS (**P* < 0.05; Figs. 5c, 6a). Additionally, we observed a high density of CD39⁺PD-1⁺ Tex cells within the TLS (*P* = 0.0210; Fig. 6b, Figs. 5d, 6a). Importantly, TCF-1 expression and CD39⁺ Tpex cell density within the TLS were respectively higher compared with the parenchyma and stroma (Fig. 6c, d), consistent with the spatial features of unsupervised CD8⁺ T cell clustering (Fig. S2E). Compared with the stroma adjacent to the TLS, the density of CD39⁺ Tpex cells inside the TLS was consistently higher than that of stromal CD39⁺ Tpex cells, irrespective of distance from the TLS (Fig. 6e). Stromal CD39⁺ Tpex cells within 400 *µm* of the TLS were significantly more concentrated in the nearest 0–100 *µm* range (**P* < 0.05; Fig. 6f). In contrast, stromal CD39⁺ dTex cells exhibited no significant difference in localization proportion between the 0–100 *µm* and 200–300 *µm* distances from the TLS (Fig. 6f). The rate of TCF-1 expression in CD39⁺PD-1⁺ Tex cells within the combined parenchyma and stroma areas was significantly higher in TLS-detectable ROIs than undetectable ROIs (*P* = 0.0338; Fig. 6g). These findings indicate that

the presence of TLSs affects the pattern of dysfunctional stages in CD39⁺PD-1⁺ Tex cells in the TME, and suggest that CD39⁺ Tpex cells are recruited via TLSs.

### CD39⁺Ki67⁺ Tex cells in blood are selectively increased 2 weeks after ICB
To better understand how TLSs contribute to the clinical benefit of ICB therapy²⁹,³⁹⁻⁴¹, it is crucial to characterize the underlying immune kinetics that shape the unique responses of CD39⁺PD-1⁺ Tex cells to ICB therapy. To determine whether ICB therapy reactivates CD39⁺PD-1⁺ Tex cells in ESCC patients, we examined changes in circulating immune dynamics before and after ICB therapy within the same research cohort (Figs. 1a, 7a–d). We analyzed 12 PBMC samples from 6 patients before and after ICB treatment using MC (Fig. 7c), applied unsupervised clustering to the PD-1⁺CD8⁺ T cell population with FlowSOM (Fig. S4A), and visualized the results using UMAP plots (Figs. 7d, S4B, S4C). This analysis identified two clusters (C1 and C4) that behaved distinctly after ICB therapy initiation (Fig. 7e). Cells in both C1 and C4 expressed high levels of CD39 and Ki67 (Fig. 7f), which is characteristic of a Tex phenotype (Fig. S4A). Further flow cytometry (FC) analysis to explore the characteristics of these clusters revealed that CD39 was predominantly expressed in the PD-1⁺ subset compared to the PD-1⁻ subset (Fig. S5C). Additionally, it was found that Ki67 was highly expressed in the CD39⁺PD-1⁺ subset compared to other subsets of CD8⁺ T cells (Fig. S5D). The exhaustion of the CD39⁺Ki67⁺ subset among PD-1⁺ T cells was confirmed both functionally and phenotypically (Figs. 8a, S5A, S5B). CD39⁺ Ki67⁺ cells in blood have been reported as neoantigen-reactive T cells in urothelial carcinoma⁴². FC analysis also confirmed a marked expansion of the CD39⁺Ki67⁺ subset following ICB (Fig. 7b), supporting a specific response of these cells to ICB therapy. Notably, the frequency of these cells was significantly higher in ESCC patients compared with healthy donors (Fig. S5E), indicating their specific presence in the TME. These findings suggest that monitoring CD39⁺Ki67⁺ Tex cells in the blood could significantly improve our understanding of the immune kinetics in ICB-treated patients.

### Proliferative CD39⁺ Tpex cells constitute a distinct subset of CD39⁺Ki67⁺ Tex cells in the blood
To examine hierarchical patterns, we focused on phenotypic and functional differences between unsupervised Tex clusters (C1 and C4) in blood (Fig. 7d, S4A). C4 expressed higher TCF-1, TOX, PD-1 and CTLA4 levels (Fig. S4A), representing proliferative CD39⁺ Tpex cells. Manual detection by MC confirmed both proliferative CD39⁺ Tpex and CD39⁺ dTex populations with a phenotypically exhausted profile expressing high levels of TOX, Eomes, PD-1, CD39, and CTLA4 (Fig. 7g, h). CD39⁺ Tpex cells in blood expressed relatively higher levels of CCR7 and CD127 (Fig. 7h) and lower levels of TIM-3 (Fig. S6A), consistent with previous reports²⁴,²⁶,³³. Despite the lower frequency of cluster C4 compared to the manually gated groups from both FC and

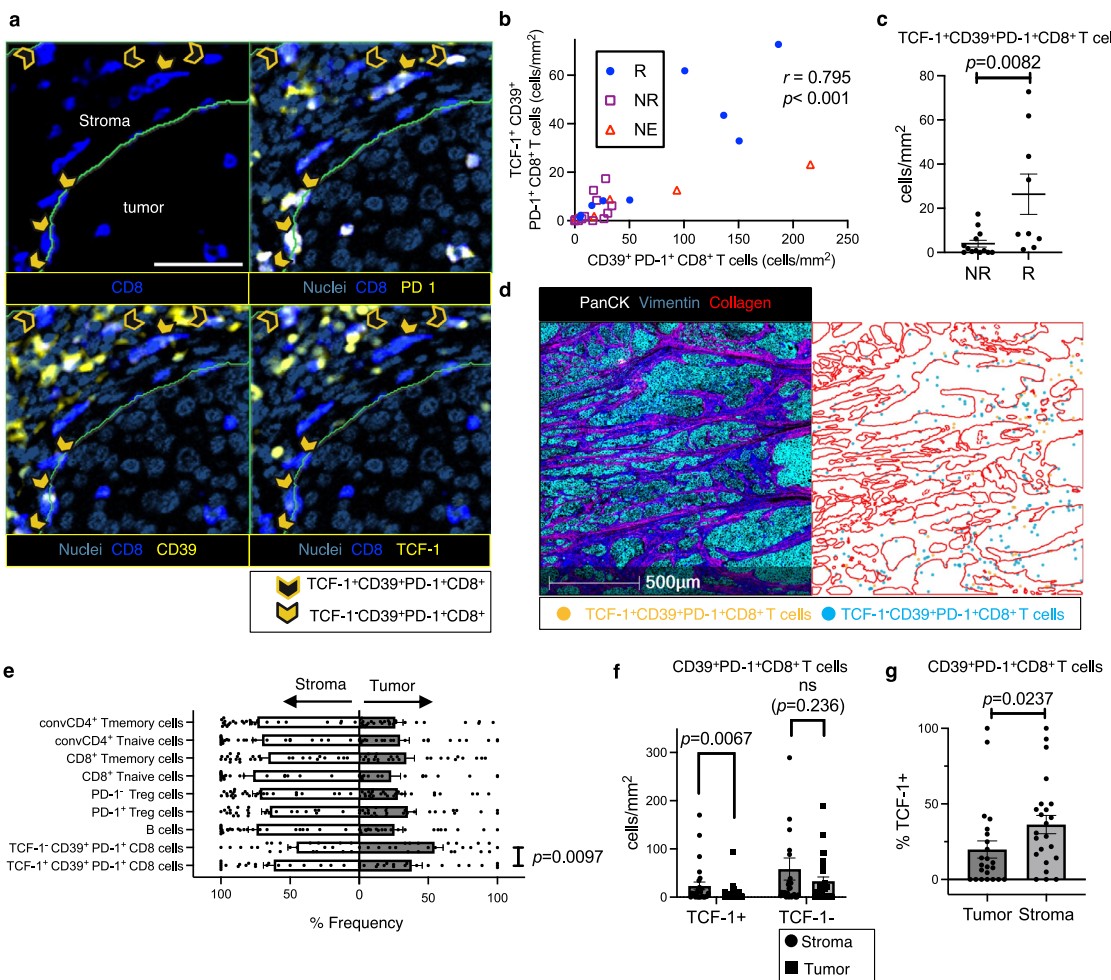

**Fig. 3 | CD39⁺PD-1⁺CD8⁺ T (CD39⁺PD-1⁺Tex) cells, forming a TCF-1⁺ subset linked to clinical outcomes of anti−PD-1 therapy, exhibit distinct spatial tumor organization reflective of their hierarchical patterns. a** IMC image of representative CD39⁺ precursor exhausted T (CD39⁺ Tpex, solid arrowheads: TCF1⁺CD39⁺PD-1⁺CD8⁺) and CD39⁺ differentiated exhausted T (CD39⁺ dTex, open arrowheads: TCF1⁻CD39⁺PD-1⁺CD8⁺) cells. The IMC images are pseudo-colored: Nuclei (dark blue), CD8 (blue), and CD39, PD-1, TCF-1 (yellow). Scale bar = 50 $\mu$m. **b** Density (cells/mm²) of CD39⁺PD-1⁺ Tex cells relative to CD39⁺ Tpex cells in the tumor (parenchyma) and stroma areas ($n = 27$); Pearson's correlations using a two-sided test ($r = 0.795$, $P < 0.0001$) (no adjustments for multiple comparisons). R: circles, NR: squares, NE: triangles. **c** Density (cells/mm²) of CD39⁺ Tpex cells in Rs ($n = 9$) vs. NRs ($n = 13$); two-sided Mann–Whitney $U$-test ($P = 0.0082$). Five NE cases are excluded from the analysis. (**d**) Spatial distribution of CD39⁺ Tpex (TCF1⁺CD39⁺PD-1⁺CD8⁺, orange) and CD39⁺ dTex (TCF1⁻CD39⁺PD-1⁺CD8⁺, blue) cells, derived from the IMC image. The red line outlines the tumor parenchyma. The

IMC images are pseudo-colored as follows: PanCK (white), Vimentin (dark blue), and Collagen (red). **e** Proportions (%) of annotated immune cells in the tumor or stroma determined from 27 patients; two-sided Wilcoxon matched-pairs signed test ($n = 21$, CD39⁺ Tpex vs. CD39⁺ dTex cells: $P = 0.0097$). Six cold tumors without CD39⁺ Tpex or CD39⁺ dTex cells are excluded from the paired analysis. **f** Density (cells/mm²) of CD39⁺ Tpex (TCF-1 + ) and CD39⁺ dTex (TCF-1-) cells in the tumor vs. stroma; two-sided Wilcoxon matched-pairs signed test (each $n = 27$, CD39⁺ Tpex: $P = 0.0067$, CD39⁺ dTex: $P = 0.236$). Stroma: circles, Tumor: squares. **g** Proportion (%) of TCF-1⁺ cells among CD39⁺PD-1⁺ Tex cells in the tumor vs. stroma; two-sided Wilcoxon matched-pairs signed test ($n = 23$, $P = 0.0237$). Four cold tumor cases lacking CD39⁺PD-1⁺ Tex cells in either area are excluded. **c, e, f, g** Error bars indicate mean ± SEM. **b, c, e, f, g** $n$ refers to independent tumor-ROIs. **a, d** Similar staining and distribution patterns were confirmed among 27 independent tumor-ROIs. **a** was created in BioRender. Kenro, T. (2024) BioRender.com/s50j163. Source data are provided as a Source Data file.

MC, a strong correlation was observed between the frequency of C4 and each manually gated group (Fig. S4D). Notably, a significant increase in C4 was observed in one particular case (Patient ID KU37), with similar results also reported in manually gated groups (Fig. 7e, S4D). Additionally, a greater proportion of CD39⁺ dTex cells than CD39⁺ Tpex cells produced IFN-γ after TCR-independent in vitro stimulation (Fig. S6C), confirming the greater cytotoxic activity of these cells[21,33]. In contrast, CD39⁺Ki67⁺ Tex cells circulating in the blood lacked CD69 expression (Fig. S6B) and resembled Tex progenitor2 or intermediately differentiated Tex cells of the four previously reported developmental stages[26,33]. These observations suggest that CD39⁺Ki67⁺ Tex populations in the blood consist of proliferative CD39⁺ Tpex cells, exhibiting phenotypic and functional features distinct from those of CD39⁺ dTex cells.

### The selective increase in circulating proliferative CD39⁺ Tpex cells following ICB therapy is associated with clinical benefit

We also investigated the association between CD39⁺Ki67⁺ Tex cells in the blood and the clinical response to ICB therapy. Following ICB therapy, the frequency of CD39⁺ dTex cells increased significantly, regardless of clinical response (Fig. 8c). However, the frequency of CD39⁺ Tpex cells showed a significant increase only in responder group ($n = 10$ patients, $P = 0.0039$; Fig. 8c). Moreover, the abundance of proliferative CD39⁺ Tpex cells at 2 weeks after ICB therapy was significantly associated with clinical response ($P = 0.0065$; Fig. 8d). The abundance (>10%) of CD39⁺ Tpex cells after ICB therapy was also significantly associated with longer progression-free survival (PFS) ($n = 28$ patients, $P = 0.0012$; Fig. 8e). These findings indicate that the increase in proliferating CD39⁺ Tpex cells in the blood following ICB

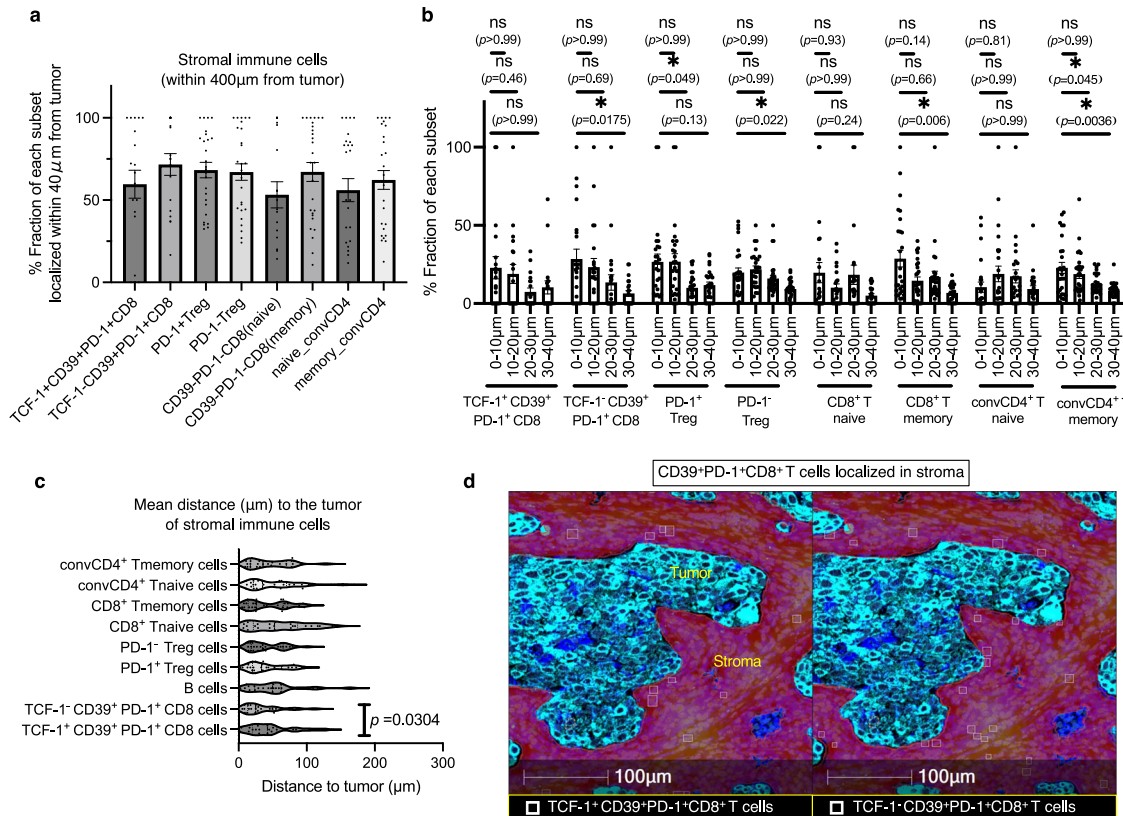

**Fig. 4 | The hierarchical exhaustion patterns of CD39⁺PD-1⁺ Tex cells lead to differences in their proximity to the tumor parenchyma. a** Proportion (%) of stromal immune cell subsets residing within 40 μm of the parenchyma among those within 400 μm of the parenchyma (tumor) from 27 independent tumor-ROIs. **b** Proportion (%) of stromal immune subsets within 400μm of the tumor parenchyma from 27 patients, compared across four distance ranges (0–10 μm, 10–20 μm, 20–30 μm, 30–40 μm) using a repeated measures analysis of variance (ANOVA), followed by post-hoc tests with Bonferroni correction. The following immune subsets were analyzed: TCF-1⁺CD39⁺PD-1⁺CD8 (*n* = 19), TCF-1⁻CD39⁺PD-1⁺CD8 (*n* = 22), PD-1⁺Treg (*n* = 26), PD-1⁻Treg (*n* = 27), CD8⁺ Tnaive (*n* = 22), CD8⁺ Tmemory (*n* = 27), CD4⁺ Tnaive (*n* = 25), CD4⁺ Tmemory (*n* = 27). Cases lacking the

respective stromal immune subset within 400μm of the tumor parenchyma are excluded from paired analysis. **c** Violin plots of median distance to the tumor of stromal immune cells from 27 patients; two-sided Wilcoxon matched-pairs signed test (*n* = 18, CD39⁺ Tpex vs. CD39⁺ dTex cells: *P* = 0.0304). Nine tumor-ROIs without stromal CD39⁺ Tpex or CD39⁺ dTex cells are excluded from the paired analysis. **d** IMC image of representative CD39⁺ Tpex and CD39⁺ dTex cells in the stroma. The cells marked with a square in each image are CD39⁺PD-1⁺ Tex cells within the red-colored stroma. The left cells are TCF-1⁺, and the right cells are TCF-1⁻. Similar distribution patterns were confirmed among 27 independent tumor-ROIs. *P* < 0.05. **b** Error bars indicate mean ± SEM. **b**, **c** *n* refers to independent tumor-ROIs. Source data are provided as a Source Data file.

therapy is correlated with clinical benefit and reflects the robust immune response driving their expansion and facilitating their subsequent differentiation into dTex cells.

### CD39⁺ Tpex cells in secondary lymphoid organs (SLOs) exhibit an exhausted profile shared with counterpart cells in the blood and tumor

Circulating tumor-specific Tpex cells are reportedly recruited from lymph nodes (LNs) into the TME[33,43,44]. Consequently, we hypothesized that CD8⁺ T cells with a CD39⁺PD-1⁺ Tex phenotype are present within SLOs. Therefore, we performed unsupervised clustering analysis of CD8⁺ T cells manually extracted from the ROIs of individually resected SLOs from three patients (Fig. S7B) and visualized these clusters using UMAP analysis (Fig. S7C). We identified clusters of CD39ʰⁱPD-1ʰⁱ Tex cells (C1–5) with high levels of TOX expression. The presence of this CD39⁺PD-1⁺ Tex subset was also confirmed in a representative SLO image (Fig. 8b). Although only a single case of HNSCC in the ICB cohort had non-metastatic LN samples available post-ICB therapy for MC analysis, we found a remarkably abundant population of CD39⁺PD-1⁺ and CD39⁺ Tpex cells within the CD8⁺ T cells (Fig. S3I). Among these clusters, C5 exhibited relatively low TCF-1 expression, possibly reflecting CD39⁺ dTex cells (Fig. S7B). However, most importantly, C1 and C2 which exhibited high levels of Ki67 expression also expressed

high levels of TCF-1 (Fig. S7B), corresponding to CD39⁺ Tpex cells. This characteristic of high Ki67 expression was similar to the findings in the dataset of resected LNs from the HNSCC that did not receive ICB treatment, where CD39⁺ Tpex cells had higher Ki67 expression rates than CD39⁺ dTex cells (Fig. S3J).

To elucidate the spatial characteristics of these clusters within LNs, we mapped each CD8⁺ T cell cluster as shown in Figure. S7A and S7D, and analyzed their proximity to major immune cells manually extracted from each ROI. We found that all CD8⁺ T cell clusters were generally in close proximity to CD4⁺ T cell subsets (Fig. S7E), compared to CD20⁺ B cells and antigen-presenting cells (CD11c⁺HLA-DR⁺). Notably in the representative SLO, CD39⁺ Tpex clusters (C1-4) were localized in significantly closer proximity to PD-1⁺ Treg cells compared to other non-exhausted CD8⁺ T cells (Fig. S7F). This proximity characteristic was consistently observed in all three cases (Fig. S7G) and suggests possible immunosuppressive conditions surrounding CD39⁺PD-1⁺ Tex cells in LNs[33,43,44].

Recently, it was reported that ICB induces the conversion of most Tpex cells in SLOs into more-differentiated Tex cells that then transit into the blood[33]. Our findings revealed that the proportion of circulating CD39⁺ Tpex cells within CD39⁺Ki67⁺ Tex cells decreased 2 weeks after ICB therapy compared to before treatment (Figs. S6D, S6E) and was further reduced by the time of disease progression (PD)

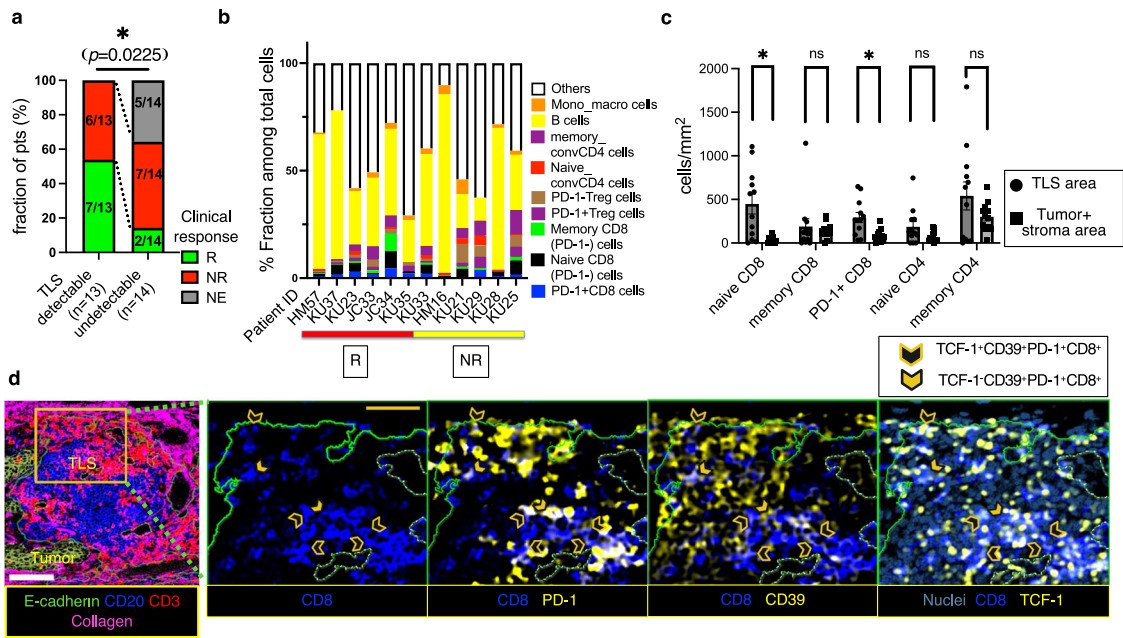

**Fig. 5 | The CD8⁺ T cell composition within TLSs associated with ICB effect is distinct from that in conventional tumor regions. a** Rate (%) of clinical response for TLS-detectable (*n* = 13 patients) vs. TLS-undetectable tumors (*n* = 14 patients); two-sided Fisher's exact test (*P* = 0.0225). **b** Fractions (%) of annotated immune cells residing in TLSs per each of 12 ROIs harboring TLSs. **c** Comparison of the density (cells/mm²) of each immune cell between TLSs (circles) and the tumor and stroma areas (squares) (*n* = 12); two-sided Wilcoxon matched-pairs signed test (each *n* = 12, naive CD8: *P* = 0.001, memory CD8: *P* = 0.569, naive CD4: *P* = 0.1099, memory CD4: *P* = 0.3804, PD-1⁺ CD8: *P* = 0.0024). **d** The representative full TLS IMC image

pseudo-colored: E-cadherin (green), CD20 (blue), CD3 (red), and Collagen (pink). The magnified images show CD39⁺ Tpex cells (open arrowheads: TCF1⁺CD39⁺PD-1⁺CD8⁺ T cells) and CD39⁺ dTex cells (solid arrowheads: TCF-1⁻CD39⁺PD-1⁺CD8⁺ T cells), pseudo-colored: Nuclei (dark blue), CD8 (blue), CD39, PD-1, TCF-1 (yellow). White scale bar = 100 μm, yellow scale bar = 50 μm. *P < 0.05. **c** Error bars indicate mean ± SEM. **c** *n* refers to independent tumor-ROIs with TLSs. **d** Similar staining and distribution patterns were confirmed among the 13 independent tumor-ROIs harboring TLSs. **d** was created in BioRender. Kenro, T. (2024) BioRender.com/s50j163. Source data are provided as a Source Data file.

(Fig. S6E). Furthermore, the increased frequency of circulating CD39⁺ dTex cells (Figs. 7e, 8c) supports the potential scenario where ICB triggers a conversion of highly proliferative CD39⁺ Tpex cells within the SLOs.

### Intratumoral CD39⁺ Tpex cells exhibit consistent high proliferative activity under ICB therapy

ICB therapy preferentially increases the frequency of circulating CD39⁺Ki67⁺ Tex cells, suggesting that the CD39⁺PD-1⁺ Tex population in the TME after ICB consists primarily of reinvigorated Tex cells recruited from the periphery[12,20]. We therefore analyzed ROIs from biopsied tumors of three cases at PD during ICB therapy (Fig. 8f). Although the sample size is small (*n* = 3 independent tumor-ROIs), the density of T cells, including CD39⁺ Tpex and CD39⁺ dTex cells, within the tumor area did not increase markedly (Fig. 8g), similar to the lack of increase in their frequency in PBMCs at PD (Fig. S6F) and consistent with previously reported findings in a study of MSI-high rectal cancer[45]. However, CD39⁺ Tpex cells exhibited consistently high Ki67 expression in the second biopsy, a pattern also observed in our IMC results prior to ICB therapy (Fig. S3B). This result is further supported by the characteristic high Ki67 expression of intratumoral CD39⁺ Tpex cells in the HNSCC dataset, which includes 6 out of 8 samples post-ICB therapy (Figs. S3B, S3C, S3D). Manual gating also confirmed that CD39⁺ Tpex cells exhibited high Ki67 expression comparable to CD39⁺ dTex cells (Figs. S3F, S3G). Furthermore, TCF-1⁺Ki67⁺CD8⁺ TILs, a key predictor of ICB efficacy in breast cancer[37], showed a significantly higher CD39⁺PD-1⁺ positivity rate compared to TCF-1⁺Ki67⁺CD8⁺ TILs in the HNSCC dataset (Fig. S3H), indicating that many highly proliferative TCF-1⁺CD8⁺ T cells are reflective of CD39⁺ Tpex cells. This characteristic aligns with the properties of CD39⁺ Tpex cells within blood (Fig. 7f) and the SLOs (Fig. S7B). Collectively, the results support the hypothesis that CD39⁺ Tpex cells in the tumor, blood, and lymphoid

tissues, characterized by high proliferative activity, are closely associated with ICB-mediated anti-tumor responses and play a critical role in therapeutic efficacy.

## Discussion

In this study, we characterize tumor-reactive CD8⁺ T cells identified by CD39 expression within the ESCC microenvironment through multiplex spatial proteomic profiling. This approach enables us to elucidate a dynamic immune landscape mediated by TLSs. To the best of our knowledge, few studies have focused on the major determinants of ICB effects using multicellular spatial analysis in ESCC.

We have revealed distinct spatial features based on the phenotypic differences of CD39⁺PD-1⁺ Tex cells, providing important insights into a possible immunodynamics mechanism. This supports the significant association between the CD39⁺ Tpex cell population and clinical benefits from ICB therapy. While Tpex cells have previously been reported to express low levels of CD39[46] or show variability in CD39 expression[26,33,34,47], a key benefit of identifying this CD39-expressing Tpex population is the confirmation of their specific tumor reactivity. Interestingly, compared to CD39⁺ dTex cells, these Tpex cells are predominantly found in the stroma and distant from the parenchyma. Conversely, CD39⁺ dTex cells are primarily localized within the parenchyma, showing a spatial bias towards this region compared to their stromal counterparts. This spatial distribution suggests a possible differentiation of Tpex cells into dTex cells during their migration into the parenchyma[38].

The dynamics of Tpex cells within TLSs have been poorly understood, and the impact of ICB on these cells has remained unclear. However, this study provides insights into the immune responses of tumor-reactive CD8⁺ T cells mediated through TLSs by elucidating the spatial characteristics of CD39⁺PD-1⁺ Tex cells around the TLSs. We observe that CD39⁺PD-1⁺ Tex cells are densely localized within the

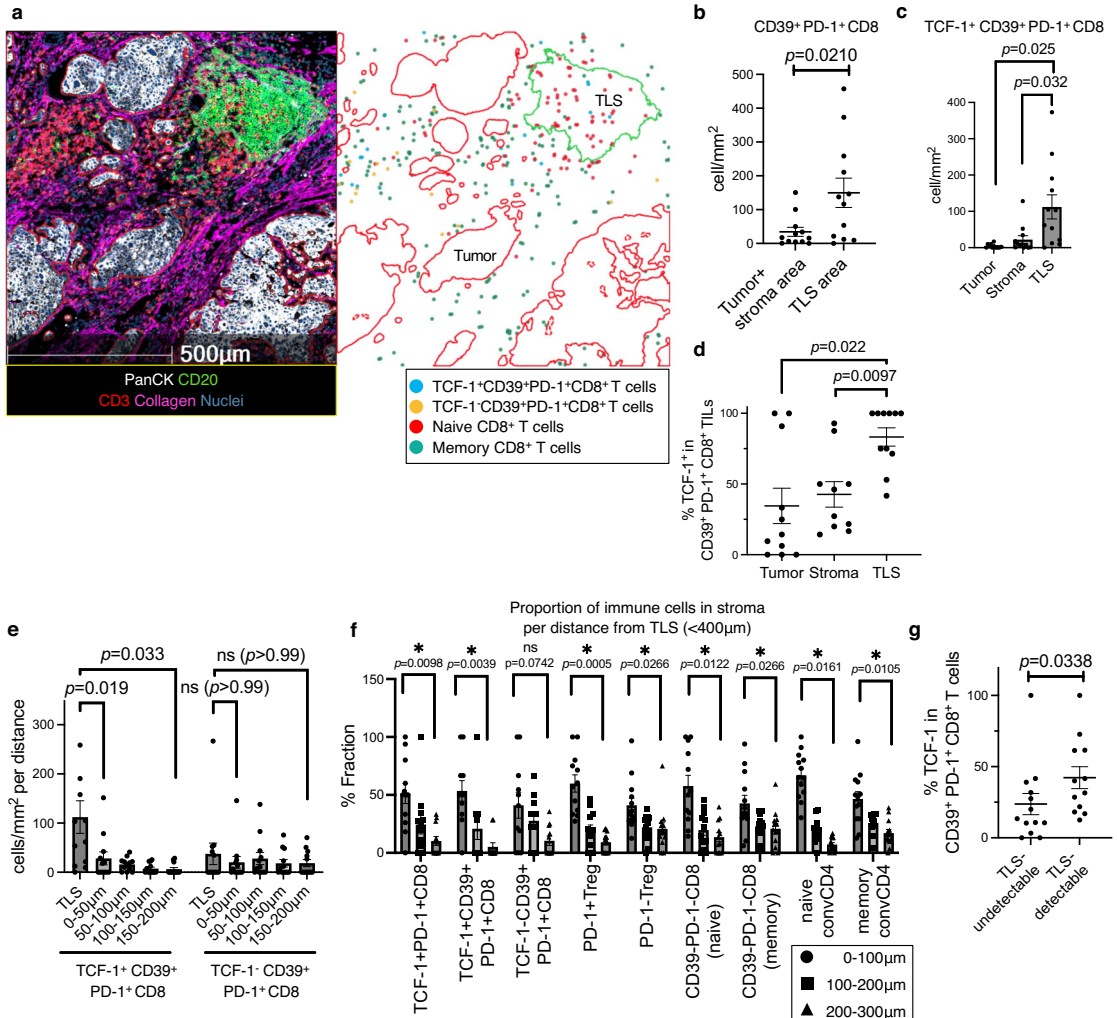

**Fig. 6 | Spatial features of CD39⁺ Tpex cells characterized by abundant localization in TLSs and near the stroma are distinct from those of CD39⁺ dTex cells.** **a** Spatial distribution of indicated CD8⁺ T cell subsets in representative IMC images with TLSs The IMC images are pseudo-colored: PanCK (white), CD20 (green), CD3 (red), Collagen (pink), and Nuclei (dark blue). **b** Density (cells/mm²) of CD39⁺PD-1⁺CD8⁺ T cells in TLSs vs. combined tumor and stroma areas; two-sided Wilcoxon matched-pairs signed test (n = 12). **c** The density (cells/mm²) of CD39⁺ Tpex cells in the tumor, stroma, and TLSs was analyzed using a repeated measures ANOVA followed by post-hoc tests with Bonferroni correction (each n = 12, TLS vs. stroma, TLS vs. tumor). **d** Fraction (%) of TCF-1⁺ cells among CD39⁺PD-1⁺CD8⁺ T cells in the tumor, stroma, and TLSs from 12 independent tumor-ROIs with TLSs; repeated measures ANOVA using a mixed-effect model followed by post-hoc tests with Bonferroni correction (TLS: n = 11 vs. stroma: n = 10, TLS: n = 11 vs. tumor: n = 11). Cases with missing data for CD39⁺PD-1⁺CD8⁺ T cells in any area (TLS, stroma, or tumor) were treated as missing data, not excluded. **e** Density (cells/mm²) of CD39⁺ Tpex and CD39⁺ dTex cells within TLSs and stromal areas at varying distances from TLSs; repeated measures ANOVA followed by post-hoc tests with Bonferroni correction (n = 12 for each). Comparisons were made between TLS and 0–50 µm, and

TLS and 150-200µm for both CD39⁺ Tpex and CD39⁺ dTex cells. **f** Proportions (%) of stromal immune cell subsets at varying distances from the TLS, relative to total cells within 400 µm of the parenchyma in 13 tumor-ROIs with TLSs. Comparisons were made among three distance ranges (0–100 µm: circles, 100–200 µm: squares, 200–300 µm: triangles) using a two-sided Wilcoxon matched-pairs signed test (0–100 µm vs. 200–300 µm). Analyzed subsets included TCF-1⁺PD-1⁺CD8 (n = 12), TCF-1⁺CD39⁺PD-1⁺CD8 (n = 11), TCF-1⁻CD39⁺PD-1⁺CD8 (n = 11), PD-1⁺Treg (n = 12), PD-1⁻Treg (n = 13), CD39⁻PD-1⁻CD8 (naive) (n = 12), CD39⁻PD-1⁻CD8 (memory) (n = 12), naive-convCD4 (n = 13), and memory-convCD4 (n = 13). Cases lacking the respective stromal immune subset within 400µm of the TLS were excluded. **g** Fraction (%) of TCF-1⁺ cells among CD39⁺PD-1⁺ Tex cells in the combined tumor and stroma areas from 25 patients; two-sided Mann-Whitney U-test (13 TLS-detectable vs. 12 TLS-undetectable tumors). Two cases with cold tumors lacking CD39⁺PD-1⁺CD8⁺ T cells are excluded. *P < 0.05. **b, c, d, e, f, g** Error bars indicate mean ± SEM. **b, c, d, e, f** n refers to independent tumor-ROIs with TLSs. **a** Similar staining and distribution patterns were confirmed among the 13 independent tumor-ROIs harboring TLSs. Source data are provided as a Source Data file.

TLSs, with a large proportion retaining the expression of TCF-1. Interestingly, we reveal distinct spatial patterns between CD39⁺ dTex cells and CD39⁺ Tpex cells in the stroma surrounding the TLSs. This supports the significant notion that TLSs might sustain the responses of CD39⁺ dTex cells to the tumor-stroma interaction produced by the CD39⁺ Tpex cells within the TLSs, replicating the function of SLOs[48]. Moreover, another recent study proposed that ICB enhances antitumor B cell responses, suggesting a potential association between TLSs and ICB responses[40,41] as well as the possibility that ICB amplifies anti-tumor Tpex responses within TLSs.

Lastly, CD39⁺PD-1⁺ Tex cells exhibit a common exhausted phenotype and proliferate in both TLSs and SLOs. Despite previous uncertainty regarding the role of CD39 expression on CD8⁺ T cells in these lymphoid tissues, it is well established that tumor-specific T cells in an exhausted state are present within SLOs[25,26,33,34,43,44]. This observation suggests that some tumor-reactive T cells express CD39, similarly to their counterparts in tumor environments. Our study shows a selective increase in CD39⁺Ki67⁺ Tex cells in the blood following PD-1 blockade, supporting the hypothesis that Tex cells expressing CD39 are recruited from SLOs to tumors[42]. Importantly, a preferential

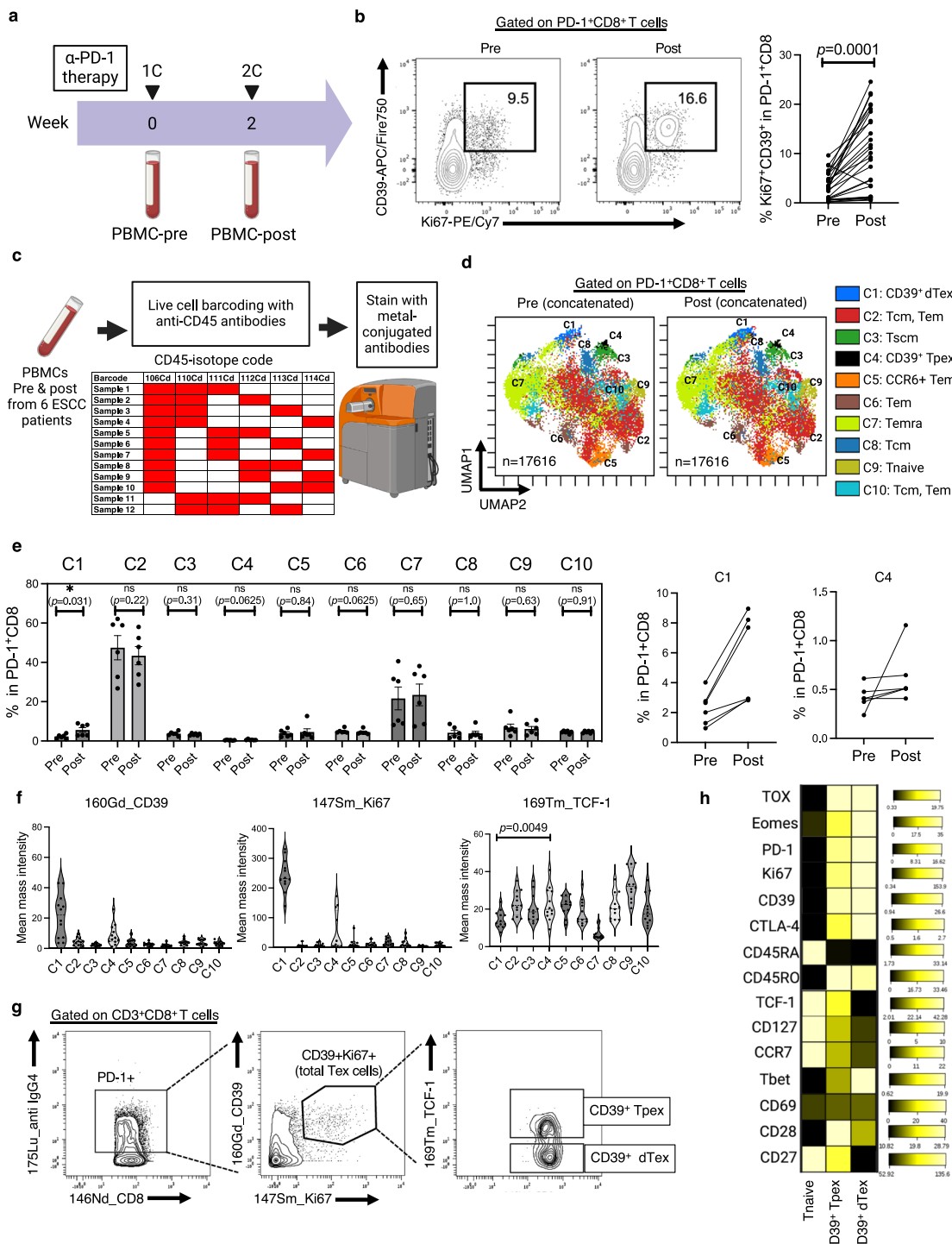

**Fig. 7 | Circulating CD39⁺Ki67⁺ Tex (CD39⁺Ki67⁺PD-1⁺CD8⁺ T) cells, partly consisting of CD39⁺ Tpex (TCF-1⁺CD39⁺Ki67⁺PD-1⁺CD8⁺ T) cells, preferentially increase in frequency 2 weeks after ICB therapy. a** Study design and treatment. **b** CD39⁺Ki67⁺ Tex cells among PD-1⁺CD8⁺ T cells pre- and post-treatment from 28 patients; two-sided Wilcoxon matched-pairs signed test ($n = 28$, pre vs. post: $P = 0.0001$). **c** Schematic illustration of MC analysis. Six patient samples were randomly analyzed. **d** Unsupervised clusters visualized in the UMAP plots for concatenated PD-1⁺CD8⁺ T cells (each 17,616 cells) from 6 patient PBMCs pre- and post-treatment. C1: CD39⁺ dTex, C2: central memory T (Tcm), effector memory T (Tem), C3: stem cell memory T (Tscm), C4: CD39⁺ Tpex, C5: CCR6⁺ effector memory T, C6: effector memory T, C7: terminally differentiated effector memory T (Temra), C8: central memory T, C9: Tnaive, C10: central memory T, effector memory T cells. **e** Comparison of the frequency (%) of 10 clusters within circulating PD-1⁺CD8⁺

T cells from six patients, analyzed by MC at pre- vs. post-treatment; two-sided Wilcoxon matched-pairs signed-rank test for each cluster (each $n = 6$). **f** Mean mass intensity of CD39, Ki67, and TCF-1 for each cluster from 12 independent PBMCs; two-sided Wilcoxon matched-pairs signed test (TCF-1, $n = 6$, C1 vs. C4: $P = 0.0049$). **g** Representative plots of CD39⁺ Tpex (TCF-1⁺CD39⁺Ki67⁺PD-1⁺CD8⁺) and CD39⁺ dTex (TCF-1⁻CD39⁺Ki67⁺PD-1⁺CD8⁺) cells determined by manual gating. Similar staining was observed across 12 independent PBMCs. (**h**) Median mass intensity marker expression of manual-gated T-naive (PD-1⁻CD27⁺CD45RA⁺), CD39⁺ Tpex, and CD39⁺ dTex cells from 12 independent PBMCs determined by heatmap analysis. *$P < 0.05$. **b, e** Error bars indicate mean ± SEM. **e, f** $n$ refers to independent PBMC samples. **a, c** were created in BioRender. Kenro, T. (2024) BioRender.com/s50j163. Source data are provided as a Source Data file.

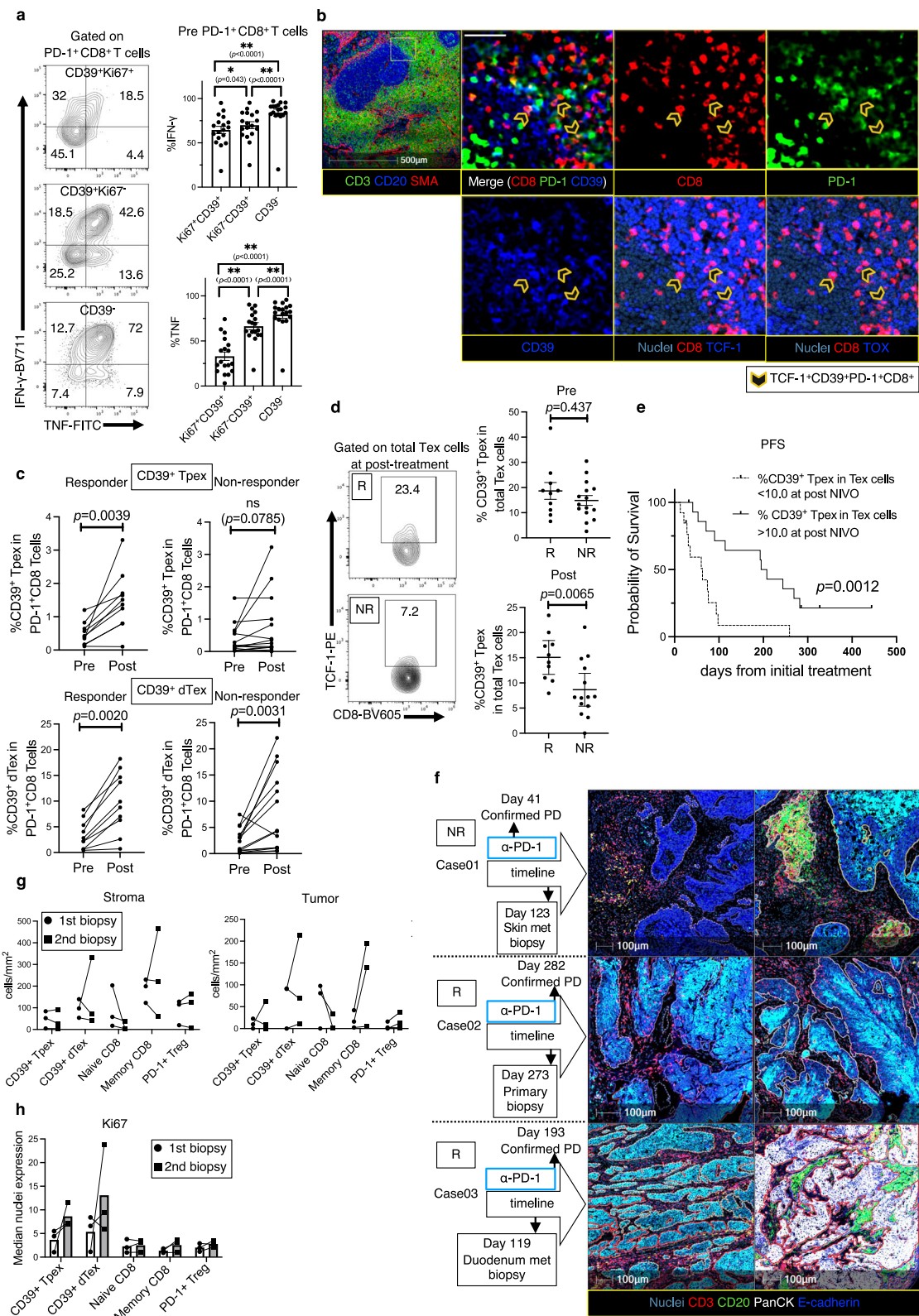

increase in CD39⁺ Tpex cells in the blood post-ICB therapy is observed only in cases of successful treatment. Recent reports indicating shared traits between Tpex cells in lymph nodes and Tex cell clones infiltrating tumors[33,34] support the notion that these CD39⁺ Tpex cells are recruited via the bloodstream. Collectively, these findings suggest that ICB-mediated anti-tumor responses can be enhanced by synergistically reinvigorating Tex cells in SLOs and TLSs.

Limitations of our study include the absence of fresh and post-treatment tumor samples, which prevents functional assessment of the tumor specificity of CD39⁺ Tpex cells. Additionally, we cannot detect all tumor-specific Tpex cells, as these cells often exhibit low CD39 expression early in exhaustion[46]. Variability in CD39 expression in Tpex cells has been reported[26,33,34,47], and our evidence regarding the tumor specificity of TCF-1⁺CD39⁻PD-1⁺CD8⁺ T cells is insufficient.

**Fig. 8 | A preferential increase in proliferative CD39$^+$ Tpex cells in the blood 2 weeks after ICB therapy is correlated with clinical benefit, and intratumoral CD39$^+$ Tpex cells consistently upregulate Ki67 expression following ICB therapy. a** IFN-γ and TNF production by indicated subsets from 18 patients prior to ICB therapy, determined by FC analysis. % IFN-γ and % TNF were compared using a repeated measures ANOVA followed by post-hoc tests with Bonferroni correction (each $n = 18$, Ki67$^+$CD39$^+$PD-1$^+$ vs. Ki67$^+$CD39$^-$PD-1$^+$ vs. CD39$^-$PD-1$^+$ T cells). Samples were randomly analyzed. **b** Representative SLO image of CD39$^+$ Tpex cells (open arrowheads). The whole SLO ROI is pseudo-colored as follows: CD3 (green), CD20 (blue), and SMA (red). In the magnified images, the colors are as follows: CD8 (red), PD-1 (green), CD39, TCF-1, TOX (blue), and nuclei (dark blue). This staining pattern is confirmed across three independent SLO samples. White scale bar = 100 μm. **c** Frequency of CD39$^+$ Tpex and CD39$^+$ dTex cells among PD-1$^+$CD8$^+$ T cells pre- vs. post-treatment from 24 patients; two-sided Wilcoxon matched-pairs signed test. CD39$^+$ Tpex (Rs: $n = 10$, NRs: $n = 14$), CD39$^+$ dTex (Rs: $n = 10$, NRs: $n = 14$). **d** % CD39$^+$ Tpex cells among CD39$^+$Ki67$^+$ Tex (total Tex) cells from 24

patients pre-treatment and post-treatment; two-sided Mann-Whitney $U$-test. Pre-treatment (Rs, $n = 10$ vs. NRs, $n = 14$), Post-treatment (Rs, $n = 10$ vs. NRs, $n = 14$). **e** Kaplan-Meier PFS for post-% CD39$^+$ Tpex cells among total Tex cells from 28 patients; log-rank test (% CD39$^+$ Tpex cells in total Tex cells > 10%, $n = 15$ vs. < 10%, $n = 13$). Two cases with missing post-treatment PBMC samples are excluded from the analysis. **f** ROIs from biopsied tumors of three patients at pre-treatment and around the time of PD on ICB therapy, pseudo-colored as follows: PanCK (white), CD20 (green), CD3 (red), E-cadherin (blue), and nuclei (dark blue). **g** Density (cells/mm$^2$) of each T cell type in the tumor and stroma pre-treatment (circles) and post-treatment (squares) from three paired tumor-ROIs. **h** Median nuclear expression of Ki67 in each intratumoral T cell type pre-treatment (circles) and post-treatment (squares) from three paired tumor-ROIs. *$P < 0.05$, **$P < 0.0001$. **a, d** Error bars indicate mean ± SEM. (**a, c, d, e**) $n$ refers to independent PBMC samples. **c, d** NE cases and those with missing PBMC samples are excluded from the analysis. **h** Box plots indicate median values. **b** was created in BioRender. Kenro, T. (2024) BioRender.com/s50j163. Source data are provided as a Source Data file.

Further research is needed on the TCR repertoire and tumor antigen reactivity of CD39$^+$ Tpex cells in tumors and blood. Moreover, the IMC analysis post-ICB therapy is constrained by a small number of cases, late time points, and the inclusion of metastatic tissues. Given the limitations of this observational study, direct evidence regarding the changes in the TME triggered by ICB therapy remains lacking. Further verification is needed to understand how CD39$^+$ Tpex cells are activated following ICB and how they contribute to antitumor activity within the TME.

In conclusion, our study reveals details related to the spatial dynamics of CD39$^+$PD-1$^+$ Tex cells in ESCC following ICB therapy. The spatial characteristics of CD39$^+$ Tpex cells underscore the significant local immune responses mediated by TLSs. A selective increase in circulating CD39$^+$ Tpex cells following ICB treatment highlights their systemic responses to ICB. Our findings could therefore contribute to the development of precision medicine approaches for cancer immunotherapy and enhance our understanding of the immunological landscape of esophageal cancer.

# Methods

## Patients

From March 2020 to July 2021, a total of 31 patients with unresectable or advanced ESCC (median age 72 years, range 40–85) were prospectively enrolled for nivolumab treatment (240 mg/patient every 2 weeks until disease progression, death, or unacceptable toxicity) (Fig. 1a). Inclusion criteria were as follows: histologically confirmed ESCC, a baseline Eastern Cooperative Oncology Group performance status of 0–1, resistance or intolerance to fluoropyrimidine-based and platinum-based chemotherapy, age over 20 years, and no prior immunotherapy (Fig. 1a, b). One patient, without the necessary chemotherapy history, was excluded from the analysis. Patient characteristics are outlined in Table S1. Written informed consent was obtained from each patient for participation in this study, and blood samples from healthy donors were collected from volunteers who also provided written informed consent. There was no additional compensation for patients. Ethical approval was centrally granted by the Ethics Committee of Kyushu University Hospital (Approval No. 2023-17) and by the respective Ethics Committees of the following institutions: Japan Community Healthcare Organization Kyushu Hospital, NHO Kyushu Medical Center, Hamanomachi Hospital, Fukuoka Wajiro Hospital, and National Kyushu Cancer Center. This study was conducted in accordance with the Declaration of Helsinki. Sex or gender were not considered in the study design as it was not relevant to our research objectives.

## Specimen collection

Blood was collected in tubes containing acid citrate dextrose solution before initial and second nivolumab administrations and at the point

of PD. PBMCs were obtained pre-treatment from all 31 patients, at 2 weeks post-ICB from 28 patients, and at PD from 19 patients (Fig. 1b). Post-ICB samples of 30 eligible patients could not be obtained from 2 patients due to patient refusal and COVID-19 restrictions. PBMCs were isolated via Ficoll gradient centrifugation and stored at −80 °C. FFPE tissue blocks were obtained from archival pathology specimens. Pre-treatment tumor samples were obtained from 27 patients, post-treatment tumor samples from 3 patients, and pre-treatment regional LNs from 3 of the 30 eligible patients (Fig. 1b). Pre-treatment FFPE samples for 3 patients were unavailable because two were held by other hospitals and one was non-tumorous. Each sample's viable tumor area was identified and mapped to its H&E slide by two pathologists (D.K. and H.Y.).

## Assessment of clinical response

Tumor response to anti−PD-1 therapy was assessed based on best response using RECIST, version 1.1, including cases without measurable lesions. Clinical benefit from ICB therapy was defined as either Non−complete response (CR)/NonPD or SD with PFS ≥ 6 months (Fig. 1c). As per prior reports[32,49], the clinical response was modified as follows: 1) patients exhibiting PR or CR were deemed clinical responders; 2) patients exhibiting NonCR/NonPD or SD with PFS ≥ 6 months were deemed responders; and 3) all other patients were classified as non-responders. Cases not evaluable (NE) included 3 patients changing treatment, including radiotherapy prior to the initial treatment evaluation, and 2 patients dying before treatment evaluation could be conducted. NE cases were excluded from the analysis based on treatment efficacy analysis.

## Antibody validation by IMC and MC

Antibodies were sourced from Standard Biotools and validated on tonsil and ESCC tissues using IMC. Antibodies were validated on PBMCs from healthy donors and ESCC patients using MC to ensure strong staining intensity and optimal signal-to-noise ratio. Unlabeled antibodies underwent metal conjugation using a Maxpar X8 Multimetal Labeling Kit (Standard Biotools). Titration and specificity were visually confirmed for new conjugates using tonsil images for IMC and PBMCs for MC. CD45-152Sm (CD45-2B11) and CD28-173Yb (EPR22076) in tumor-ROIs exhibited an extremely low signal-to-noise ratio and was therefore not used for analysis. Further antibody and critical reagents are described in Tables S2, S3, S4 and S5.

## IMC staining procedure

Serial sections (4 $\mu$m) from ESCC patient FFPE blocks were prepared and stained per the Standard Biotools IMC protocol. The tumor samples were de-waxed by baking at 60 °C for 2 h, followed by deparaffinization in fresh xylene (twice, 10 min each) and rehydration

through a graded alcohol series (5 min each at 100%, 95%, 80%, 70%). Antigen retrieval was performed using a decloaking chamber with Tris-EDTA buffer (Antigen Unmasking Solution, pH 9.0, Vector Laboratories) at 110 °C for 20 min, followed by cooling to 70 °C at room temperature (RT). Slides were blocked with 3% BSA in Maxpar PBS at RT for 60 min. Concurrently, the antibody cocktail, prepared in 0.5% BSA buffer, was applied post-blocking and incubated overnight at 4 °C. Following incubation, slides were washed and treated with Cell-ID Intercalator-Ir (Standard Biotools) for DNA staining at RT for 30 min. Finally, slides were rinsed and air-dried for at least 20 min before IMC acquisition.

## IHC staining procedure

IHC staining was performed on serial tumor sections using the following antibodies: CD3 (clone F7.2.38, Agilent Technologies), CD20 (clone L26, Agilent Technologies), and anti-PNA (MECA79R, Novus). The slides were first deparaffinized and rehydrated, followed by antigen retrieval using EDTA buffer (Antigen Retrieval Buffer, EDTA buffer, pH 8.0, Abcam) through a 20-min microwave treatment. Endogenous peroxidase activity was blocked by incubating the slides in 3% hydrogen peroxidase for 5 min. Protein blocking was performed using Protein Block Serum-Free (Agilent Technologies). After applying the primary antibody, a DAKO Envision Detection System (Agilent Technologies) was used for primary antibody detection.

## Selection of ROIs for IMC analysis

Whole immunohistochemical-stained slides from all cases were scanned to identify areas rich in CD3+ cells and CD3+/CD20+ TLSs within the tumor. For tumors lacking TLS, areas with a high density of CD3+ cells were randomly mapped from consecutive IHC sections stained for CD3. One ROI per mapped area was then acquired from IMC-stained samples (Fig. S8A). Before the acquisition of ROIs within TLSs, TLSs were initially screened in tumor-adjacent or -inherent areas using consecutive IHC sections stained for CD3 and CD20. Subsequently, randomly selected TLSs were then mapped to their corresponding locations on IMC-stained samples, ensuring that at least one tumor site per TLS was included. The accuracy of acquiring designated ROIs was experimentally confirmed twice using the representative sample (patient ID KU37) (Fig. 2b). Upon verification, for each sample in which TLSs were detectable, ROIs corresponding to TLSs and associated tumor areas were captured (Fig. S8C). Optionally, additional tumor ROI without TLSs, previously mapped based on IHC-stained serial sections, were also acquired. Given the predefined conditions described above, the area for ROI acquisition was predetermined, and ROIs were acquired within a range of up to 2.25 mm² (0.39–2.25 mm²).

## IMC data acquisition

IMC images were acquired using a Hyperion Imaging System (Standard Biotools) linked to a Helios mass cytometer (Standard Biotools) (Fig. S10A). All images were collected according to the manufacturer's guidelines. Following daily calibration, each tissue slide was subjected to 200-Hz laser ablation. The resulting IMC data were saved as MCD files. The integrity and quality of staining across all samples in each channel were manually verified by reviewing the visualized MCD files in MCD Viewer (ver. 1.0.560.6, Standard Biotools). Figure S9A and S9B show the expression of all markers in SLOs or tumors.

## IMC image analysis of tumor and SLO samples

Raw data from 27 eligible patients were converted to 32-bit TIFF via MCD Viewer, followed by processing using two different imaging methods (Fig. S10B, S10C). Briefly, while both methods were applied to tumor samples, "IMC image processing B", using HALO image analysis software (ver. 3.5.3577.265, Indica Labs), was predominantly employed for these samples, as per the manufacturer's protocol. "IMC image processing A", based on a previously reported algorithm[50], was utilized

to generate spatial information for CD8+ T cell clustering within tumors. For SLOs, "IMC image processing A" was primarily utilized due to its superior small lymphocyte nuclei identification capability compared to image processing B.

## IMC image denoising

We noted the presence of hot pixels and shot noise in the IMC images obtained for several markers, which could potentially affect downstream analysis. To clearly visualize protein expression, particularly in channels with low signal-to-noise ratios, set precise positive thresholds, and improve cell clustering, we utilized the IMC-Denoise software package[51]. This software was used to analyze markers including CD3, CD4, CD8, CD20, PD-1, CD39, TOX, Ki67, CD45RA, CD45RO, HLA-DR, CD11c, FoxP3, TCF-1, LAG-3, OX-40, TIM-3, CD69, and CD25.

IMC-Denoise employs two main steps: a differential intensity map-based restoration (DIMR) for hot pixel removal and a self-supervised deep learning algorithm for shot noise image filtering (DeepSNiF) for shot noise suppression. DeepSNiF requires model training prior to implementation. We trained a separate model for each marker, resulting in the creation of a separate dataset for each marker from our IMC images after DIMR hot pixel removal. Each dataset comprised ~30,000 64 × 64 patches after data augmentation. Subsequently, all models were trained on a single NVIDIA RTX 3000 GPU over 200 epochs, using the default parameter settings of the software. After training, the DIMR-processed IMC images were denoised by DeepSNiF using their corresponding trained models. The denoised IMC images were then evaluated by a pathologist (T.Iw.) (Fig. S11A). The quality of these images was confirmed to be comparable in quality to immunofluorescence staining for CD39, PD-1, and CD8 (Figs. S11B, S11C).

## IMC image processing A for tumor and SLO samples

**Pixel classification.** Pixel classification training was performed using ilastik software (ver. 1.3.3 post1)[52] to enhance the quality of cell identification in Cellprofiler (ver. 4.2.1)[53] (Fig. S10D). The following channels of raw data were selected for machine learning: 196Pt-ICSK2, 191Ir-DNA, 193Ir-DNA, 156Gd-CD4, 161Dy-CD8, and 162Dy-CD20, which were used to train the pixel classifier. Batch processing for pixel classification was not undertaken. In some instances, pixel classification retraining was necessary to minimize uncertainty and enhance the quality of the pixel classification results. Both training and probability map generation were performed in ilastik, with all features selected having sigma values ranging from 0.3 and 1.6. The resultant probability maps were saved as individual 16-bit TIFF files for each class.

**Single-cell segmentation.** The generated probability maps were processed using Cellprofiler (ver. 4.2.1) to create nuclei and membrane masks. An Otsu three-class thresholding approach was used to segment nuclear masks by overlaying both 191Ir and 193Ir-DNA maps (Fig. S10D). Nuclei were identified as primary objects, with a specified diameter ranging between 3 and 18 pixels. Subsequently, membrane masks were created by overlaying the probability maps of 196Pt-ICSK2, 156Gd-CD4, 161Dy-CD8, and 162Dy-CD20. The intensity of each overlaid channel was adjusted for each sample, thereby enhancing the detection of CD4+, CD8+, and CD20+ cells. Using the propagation method, secondary objects were identified by overlaying the nuclei and membrane masks (Fig. S10E). Cell mask images for single-cell segmentation were subsequently generated and saved for further analysis. The quality of cell segmentation was verified using histoCAT (ver. 1.761)[54], and single-cell measurements were extracted from all available channels using the mean pixel values for each segmented cell. Representative images of cell-segmented tumor samples and SLO samples are shown in Figure S10F.

**Single cell data processing.** FCS formats from ROIs were processed using Cytobank (https://cytobank.org). Spillover correction was

initially performed by referring to a previously reported spillover matrix[55]. If residual spillover was observed in samples used for antibody titration, manual adjustments to the spillover matrix were performed. Subsequently, cell data were normalized to the 99th percentile and then subjected to Z-score normalization[51,56].

**Gating strategy of cell subsets.** For tumor-ROIs (Fig. S12C), manually gated subsets including CD8$^+$ T cells, epithelial cells, and CD20$^+$ B cells were extracted for further analysis. Similarly, for SLO-ROIs (Fig. S12D), the manually gated subsets, as described in Figure S12A, were also extracted.

**High-dimensional data analysis of IMC tumor data.** FlowSOM and UMAP analyzes were performed using Cytobank (https://cytobank.org). To ensure a sufficient number of cells for Tex phenotype verification, FlowSOM analysis[33,56] was performed using equal sampling from each of the 11 ROIs, each containing over 500 CD8$^+$ T cells. The analysis identified 225 clusters and 7 metaclusters specifically for CD8$^+$ T cells. Key markers used to identify the Tex clusters included OX40, LAG3, TIM3, TCF1, TOX, PD-1, CD45RA, CD69, Ki67, CD39, and CD25 with an arcsinh cofactor of 1 for parameter transformation. For visualizing the phenotype, the resulting clusters were analyzed using UMAP, applying the same parameters as in FlowSOM, with settings adjusted to 15 neighbors and a minimum distance of 0.01.

**High-dimensional data analysis of IMC SLO data.** CD8$^+$ T cells extracted from SLOs were analyzed separately due to the distinct panels used for tumors. All cells from three samples were included in the downstream analysis, reflecting the abundance of T cells. For each SOM, 225 clusters and 10 metaclusters were identified for CD8$^+$ T cells. The following markers were used to identify the Tex clusters: OX40, LAG3, TIM3, TCF1, TOX, PD-1, CD45RA, CD69, Ki67, CD39, CD25, CD45RO, and HLA-DR with an arcsinh cofactor of 1 for parameter transformation. For phenotype visualization, the resulting clusters were analyzed using UMAP, employing the same parameters as in FlowSOM, with the algorithm settings adjusted to 15 neighbors and a minimum distance of 0.01.

**Spatial analysis using CytoMAP.** To visualize the distribution of cell populations, CytoMAP analysis was employed[57]. Within SLOs, the "Calculate Distance" function was used to assess the proximity of CD8$^+$ T cell clusters to manually gated immune cells (Fig. S12D). The "Make surface" function was then used to delineate the manually annotated regions of TLSs and tumors (Fig. S2D). For TLS annotation, manually gated CD8$^+$ T cells, CD4$^+$ T cells, and CD20$^+$ B cells were used, while epithelial cells were used to annotate the tumor parenchyma (Fig. S12C).

**IMC image processing B for tumor samples**
**Visualization and HALO software.** For image processing B, 32-bit TIFF files containing the denoised images were loaded into HALO software. These images were visualized by manually adjusting the threshold value of each marker's expression, followed by a review and additional adjustments by the pathologist (T.I.), who was blinded to the clinical outcomes, to refine cell phenotyping (Fig. S12B). Images of tumor samples, highlighted by representative markers, are shown on the left side of Figure S8A and S8C. Due to batch effects among individual cases, each analysis using HALO modules was processed separately for each dataset. The following modules were used: 1) Highplex FL, 2) Tissue Classifier, and 3) Spatial Analysis.

**Image processing with the highplex FL module.** With the Highplex FL module, cell segmentation was primarily conducted at the nuclear and cell membrane levels, followed by cell phenotyping. For the initial identification of cell nuclei, overlays of 191Ir-DNA and 193Ir-DNA were used. Subsequently, to construct cell membrane outlines around the nuclei, we used 156Gd-CD4, 162Dy-CD8, 161Dy-CD20, and a Maxpar IMC Cell Segmentation kit (195Pt-ICSK1, 196Pt-ICSK2, 198Pt-ICSK3) (Standard Biotools) for cell membrane detection. The thresholds for positivity of each marker's expression were manually determined for cell phenotyping, with adjustments made in increments of 0.1 to 0.2 to obtain the optimal threshold. These thresholds were finalized following review by a pathologist (T.Iw.), who based decisions on the individual staining characteristics of nuclei or membranes, thereby confirming cell phenotypes defined by the distinctive characteristics of each marker's expression (Fig. S12B). Specifically, for threshold setting in low signal-to-noise channels such as CD39 and PD-1, there was a strong correlation ($r = 0.827$, $P < 0.0001$) between the cell density of CD39$^+$PD-1$^+$ CD8$^+$ T cells identified through automated gating in Image processing A and the visual threshold setting by HALO, confirming sensible cut-off selection (Fig. S12E). An extensive image displaying various cell phenotypes after processing with the Highplex FL module is shown in Figure S8B. For the heatmap analysis of the T cell subset, processed data were normalized to the 99$^{th}$ percentile followed by Z-score normalization. Ki67 expression was evaluated using the raw signal for each subset.

**Image processing with the tissue classifier module.** Subsequent to cell phenotyping, tissue classification across the entire ROI was performed using the Random Forest algorithm within the Tissue Classifier module implemented in HALO. Within the tumor area, five types were classified: tumor parenchyma, tumor stroma, portions of the normal region, TLSs, and blank regions. The training and prediction steps in this module were performed individually for each ROI. The markers used for these tissue classifications included E-cadherin, Pan-cytokeratin (Pan-CK), Collagen I, αSMA, vimentin, the nuclear markers 191Ir and 193Ir-DNA, CD3, and CD20. Images processed using the Highplex FL module, showcasing delineated boundary lines between the parenchyma and stroma, are shown on the right side of Figure S8A. Additionally, annotations for TLSs are provided on the right side of Figure S8C.

**Image processing using the Spatial Analysis module.** To integrate cell phenotype with spatial information, the Spatial Analysis module implemented in HALO was used to calculate the distances and the density of each cell phenotype per unit distance, specifically analyzing how these cells are distributed within the stroma relative to the tumor parenchyma or TLSs.

**MC staining and data acquisition procedure**
Staining and data acquisition were carried out in accordance with the manufacturer's guidelines. Notably, Cd-CD45 antibody barcoding mixtures were created according to the barcode key (Fig. 7c) and then added to the 12 corresponding sample tubes, each containing $2 \times 10^6$ cells. In addition, PD-1 expression on CD8$^+$ T cells in peripheral blood was analyzed using 175Lu anti-human IgG4 (clone HP6025, Southern Biotec) following saturation with nivolumab (human IgG4, Sellek Chemicals). Figure S4F presents a comparison of the direct and indirect methods for detecting PD-1 expression by MC.

**Detailed MC staining procedure**
A solution was prepared using RPMI 1640 medium supplemented with 10% FBS, 5 mM MgCl$_2$, and 15 units/ml of DNAase I and incubated at 37 °C in a 5% CO$_2$ environment. Cryopreserved PBMCs were thawed in a 37 °C water bath, transferred to a 5-ml tube containing the aforementioned solution, and centrifuged at $300\,g$ for 10 min. After removing the supernatant by aspiration, an additional 3–4 ml of the same solution was added, followed by incubation at 37 °C for 30 min. The sample was then centrifuged at $300\,g$ and the supernatant was removed by aspiration. The pellet was washed twice using Maxpar Cell Stain Buffer (CSB) (Standard Biotools) supplemented with 1 mM EDTA.

Fc-blocker was added, followed by a 10-min incubation at RT, and the cells were then washed with CSB plus 1 mM EDTA.

After the supernatant was removed by aspiration, the cells were incubated for 30 min at RT with a solution containing Nivolumab at 25 μg/ml and 103Rh DNA intercalator (1:500) (Standard Biotools). After washing, the cells were incubated at RT for 30 min in CSB with 175Lu anti-IgG4. After two more washes, the samples were incubated for 30 min at RT with live-cell barcoding antibody mix against CD45. After four washes with CSB plus 1 mM EDTA, all 12 samples were mixed and then incubated for 30 min at RT with a surface antibody cocktail. The cells were then washed once with CSB plus 1 mM EDTA and incubated for 30 min at RT with nuclear antigen staining buffer concentrate (4×): nuclear antigen staining buffer diluent mixed at a 1:3 ratio (Standard Biotools). After washing twice with 2 ml of Maxpar antigen staining Perm (Standard Biotools), intracellular antibody mix was added, and the solution was incubated for another 30 min at RT. After washing once with 2 ml of Maxpar antigen staining Perm, the cells were fixed with a 1.6% FA solution for 15 min. Following a wash with Maxpar antigen staining Perm, the cells were incubated for 2 h at 4 °C in Maxpar Fix and Perm buffer solution containing Cell-ID Intercalator-Ir (1:1000; Standard Biotools). After two washes with CSB, the cells were washed twice with Maxpar Cell Acquisition Solution (CAS) (Standard Biotools). The supernatant was then removed by aspiration, and the cells were resuspended in a solution of EQ Four Element Calibration Beads (Standard Biotools) to CAS at a 1:9 ratio, followed by analysis using a Helios mass cytometer (Standard Biotools).

## MC data acquisition
Prior to analysis, each barcoded MC sample was adjusted to ~1 × 10⁶ cells and exposed to CAS. The samples were then analyzed at an event rate of 300–400 events/s.

## Data normalization and de-barcoding for MC
The original FCS files were normalized and subsequently de-barcoded using CyTOF software (ver. 7.0.8493, Standard Biotools) according to the manufacturer's instructions. Each of the 12 FCS files, extracted as normalized and debarcoded CD45⁺ cells, was saved and prepared for further analysis using Cytobank.

## MC clustering
MC clustering was performed based on a previously described method using Cytobank[58]. CD8⁺ T cells were manually gated as CD3⁺CD8⁺CD4⁻ cells. Manual gating for PD-1 was also performed, as shown in Figure S4F. UMAP analysis was then applied to manually gated PD-1⁺CD8⁺ T cells from each of the 12 samples, involving a total of 17,616 cells equally sampled across all cases. The following markers were used to generate the UMAP plots: CCR6, CD45RA, CD4, Ki67, OX40, CD28, TIM3, TOX, CXCR3, CD27, CD39, Tbet, CD69, CD45RO, LAG3, CCR7, CD127, TCF1, CTLA4, CXCR5, CX3CR1, CXCR4, Eomes, and PD-1. Parameters were set to 30 neighbors and a minimum distance of 0.01. The resulting UMAP plots were fed into FlowSOM clustering with algorithm settings of 100 clusters and 10 metaclusters.

## High-dimensional data analysis of MC dataset
FlowSOM and UMAP analyzes were performed on eight available tumor samples, obtained through biopsy or resection, from the ICB-treated cohort of HNSCC patients[33] using Cytobank. CD8⁺ T cells were manually gated, as described in Figure S3I. FlowSOM analysis employed equal sampling for consistency across samples. For each SOM, 225 clusters and 9 metaclusters were identified for CD8⁺ T cells. The markers used for this analysis included PD-1, CD39, TCF-1, Ki67, TIM-3, CD127, CD69, CD103, GranzymeB, Tbet, CD39, CD38, CD25, CD45RA, CCR7, CD27, TCF1, TIGIT, and HLA-DR. The resulting clusters were then analyzed using UMAP, employing the same parameters as in

FlowSOM with the algorithm adjusted for 15 neighbors and a minimum distance of 0.01.

## Cell quantification of MC dataset
Quantification by manual gating was performed on 8 tumor samples, 1 non-metastatic lymph node (LN) sample from the dataset of the ICB cohort of HNSCC patients, and 9 non-metastatic LN samples from the standard of care cohort[33]. The expression rates of TCF-1, PD-1, CD39, and Ki67 in CD8⁺ T cells were determined based on the gating strategy shown in Figure S3I.

## Flow cytometry analysis
PBMCs collected pre-treatment and during nivolumab treatment were thawed and stained using a Zombie Aqua Fixable Viability kit (BioLegend). Cells were then stained with fluorochrome-conjugated antibodies against surface proteins of interest. For intracellular staining, the cells were fixed and permeabilized using Fixation/Permeabilization concentrate and Fixation/Permeabilization diluent (Thermo Fisher Scientific) and stained with fluorochrome-conjugated antibodies against intracellular proteins. PD-1 was detected using anti-human IgG4-Alexa Fluor 647 (HP6025, Southern Biotec) after saturation with nivolumab (Fig. S4E). The following markers were used with an Attune flow cytometer (Thermo Fisher Scientific): Ki-67 (BioLegend, Ki67), CD4 (BioLegend, SK3), CD39 (BioLegend, A1), Eomes (eBioscience, WD1928), Tox (eBioscience, TXRX10), Tbet (BioLegend, 4B10), CD14 (BioLegend, M5E2), CD8 (BioLegend, SK1), TIM-3 (BD, 7D3), CD69 (BioLegend, FN50), TCF-1 (BioLegend, 7F11A10), OX-40 (BioLegend, ACT35), IFN-γ (BD, 4S.B3), and TNF (BD, Mab11). Gating for all markers except lineage markers was determined based on isotype controls, with a representative gating strategy shown in Figure S6G. Further antibody details are described in Table S5.

## Cytokine analysis
Thawed cells were stimulated with phorbol 12-myristate 13-acetate (Sigma) at 0.25 μg/ml and ionomycin (Sigma) at 2.5 μg/ml for 3 h in a 37 °C incubator and then stained. Cytokine production was analyzed by intracellular staining using antibodies against IFN-γ (BD, 4S.B3) and TNF (BD, Mab11). A sample from an ineligible patient was used in this analysis.

## Immunohistochemical analysis of PD-L1
PD-L1 staining of samples from each eligible patient was performed using FFPE tumor samples and an anti−PD-L1 antibody (1:100, 28-8, Abcam). The CPS was calculated by a pathologist (K.D.).

## Immunofluorescence Staining
Fluorescence multiplex immunostaining was performed on FFPE samples from an enrolled patient to assess the staining quality of denoised images. Antigen retrieval was performed using Antigen Retrieval Solution pH 9 (Nichirei, Tokyo, Japan) and microwaving for 20 min. The primary antibodies used were as follows: PD-1 (Clone NAT105, ab52587, Abcam), CD39 (Clone ERP20627, ab223842, Abcam), and CD8 (Clone 4B11, PA0183, Leica Biosystems). All primary antibodies were incubated at RT for 90 min. The following secondary antibodies were used: anti-mouse IgG2b CF Dye 647 (Biotium), anti-mouse IgG1 CF Dye 488 (Biotium), and anti-rabbit IgG CF Dye 568 (Biotium). All secondary antibodies were incubated at RT for 40 min. DAPI was used as the nuclear counterstain. The images were captured using a Leica DMi8 THUNDER imaging system (Leica, Weztlar, Germany).

## Statistical analysis
Statistical analyzes for group comparisons and correlations were conducted using the Prism 9.5.1 and JMP pro 16 software packages.

Data normalization and denoising of IMC images were performed in Python 3.6. The treatment effect difference between the two TLS-based groups was compared using a two-sided Fisher's exact test. The relationships of continuous variables between two groups were analyzed using non-parametric Mann–Whitney $U$-tests or Wilcoxon matched-pairs signed rank tests for two-sided unpaired and paired analyzes, respectively. For multiple groups, repeated measures ANOVA with post-hoc tests using Bonferroni correction (including one model using a mixed-effect model) or a Kruskal-Wallis test with post-hoc Dunn comparisons were used. Correlations between continuous variables were determined by Pearson's r coefficient using a two-sided test. For all tests, $P < 0.05$ was considered statistically significant. PFS was defined as the time from the initiation of ICB therapy to the date of confirmed PD or death from any cause. The cutoff value for the frequency of CD39+ Tpex cells was determined using ROC curve analysis. The optimal threshold in which the sum of sensitivity and specificity was the maximum was detected.

### Reporting summary

Further information on research design is available in the Nature Portfolio Reporting Summary linked to this article.

## Data availability

The MC and IMC data generated in this study are publicly available in the Zenodo database under the accession code https://doi.org/10.5281/zenodo.11421121 for academic non-commercial research[59]. The MC and IMC data generated in this study are also provided in the Supplementary Information/Source Data file. The HNSCC MC dataset[33] can be downloaded from https://doi.org/10.17632/2zgppyr2rr.1. All remaining data are included in the Supplementary Information or available from the authors. The raw numbers for charts and graphs are available in the Source Data file whenever possible. Source data are provided with this paper.

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

## Acknowledgements

This work was supported by the Japan Society for the Promotion of Science, Grant-in-Aid for Scientific Research (grant numbers 20K08311 and 23K07376) awarded to E.B. We extend gratitude to the patients and volunteers who participated in this study. We acknowledge Hiroshi Fujii for providing pathological specimens, and we appreciate the assistance and contributions from the Kyushu Medical Oncology Group (KMOG), specifically Satoshi Nishiyori, Sakuya Koreishi, Ryosuke Taguchi, Satoru Yamaga, Kohei Arimizu, Takashi Imajima, Takafumi Kitazono, Aya Takigawa, Shohei Ueno, Kenta Nio, Yuzo Matsushita, Tomoyasu Yoshihiro, Kosuke Sagara, Kanami Furukawa, Miyuki Kuwayama, Yasumune Doi, Masato Komoda, Fumiyasu Hanamura, and Yuta Okumura, as well as Yuqing Wang from the Department of Medicine and Biosystemic Science at Kyushu University Faculty of Medicine, Kyushu University, for their invaluable contributions. All graphical images were created using BioRender.com. The authors thank FORTE Science Communications for English language editing.

## Author contributions

K.Ta. and E.B. designed the study. K.Ta. performed experiments and analyzed all data. K.Ta., H.O., K.U., P.L., and T.Iw. assisted with the experiments and analysis. D.K., T.Iw., and H.Y. examined FFPE samples. Yu.S., S.T., H.S., H.A., Yo.S., R.T., H.K., T.E., and K.M. treated the enrolled patients and collected the clinical data. K.Ta., H.O., K.U., M.I., K.Y., K.Ts., P.L., T.Is., H.K., D.K., T.Iw., H.Y., Y.O., K.A., and E.B. discussed the experimental data. K.Ta., H.O., K.U., P.L., K.Y., T. Iw., and E.B. wrote the manuscript. All authors reviewed and participated in the revision of the manuscript.

## Competing interests

K.Ta. received honoraria for lectures from Ono Pharmaceutical. M.I. received honoraria for lectures from Ono Pharmaceutical. K.Y. received honoraria for lectures from Ono Pharmaceutical. K.Ts. received honoraria for lectures from Ono Pharmaceutical. Yu. S. received honoraria for lectures from Ono Pharmaceutical. S.T. received honoraria for lectures from Ono Pharmaceutical and Bristol-Myers Squibb. H.S. received honoraria for lectures from Ono Pharmaceutical. H.A. received honoraria for lectures from Ono Pharmaceutical and Bristol-Myers Squibb. T.E. received honoraria for lectures from Bristol-Myers Squibb and Ono Pharmaceutical, and research grants from Ono Pharmaceutical outside of this study. K.A. received honoraria for lectures from Bristol-Myers Squibb and Ono Pharmaceutical, and research grants from Bristol-Myers Squibb and Ono Pharmaceutical outside of this study. E.B. received honoraria for lectures from Bristol-Myers Squibb and Ono Pharmaceutical. The other authors declare no competing interests.

## Additional information

[1]Department of Medicine and Biosystemic Science, Graduate School of Medical Sciences, Kyushu University, Fukuoka, Japan. [2]Department of Oncology and Social Medicine, Graduate School of Medical Sciences, Kyushu University, Fukuoka, Japan. [3]Department of Hematology/Oncology, Japan Community Healthcare Organization Kyushu Hospital, Fukuoka, Japan. [4]Department of Imaging Science Program, McKelvey School of Engineering, Washington University in St. Louis, St. Louis, MO, USA. [5]Department of Medical Oncology, NHO National Hospital Organization Kyushu Medical Center, Fukuoka, Japan. [6]Department of Medical Oncology, Hamanomachi Hospital, Fukuoka, Japan. [7]Department of Medical Oncology, Kitakyushu Municipal Medical Center, Fukuoka, Japan. [8]Department of Medical Oncology, Fukuoka Wajiro Hospital, Fukuoka, Japan. [9]Department of Medical Oncology, St Mary's Hospital, Kurume, Japan. [10]Department of Gastrointestinal and Medical Oncology, National Kyushu Cancer Center, Fukuoka, Japan. [11]Department of Medical Oncology, Sasebo Kyosai Hospital, Nagasaki, Japan. [12]Department of Anatomic Pathology, Graduate School of Medical Sciences, Kyushu University, Fukuoka, Japan. [13]Department of Pathology, Graduate School of Medicine, Dentistry and Pharmaceutical Sciences, Okayama University, Okayama, Japan. ✉e-mail: baba.eishi.889@m.kyushu-u.ac.jp

