## [Peer Review File · Nature Communications]

Spatial dynamics of CD39⁺CD8⁺ exhausted T cells reveal tertiary lymphoid structures-mediated response to PD-1 blockade in esophageal cancerREVIEWER COMMENTS

Reviewer #1 (IMC)(Remarks to the Author):

The authors present an interesting and in depth study of the spatial distribution of immune subsets in esophageal cancer, with a focus on the exhausted T-cell subsets. Overall, the manuscript is impactful and readable. This reviewer has some issues that could be addressed by the authors that focus on methodological presentation and details, as well as on some of the results.

Methods:

Overview comment: the description the computational and imaging methods in the main manuscript is very short. The methods should be described in detail in the main manuscript rather than the supplementary materials as the approach here is still somewhat novel (leveraging IMC).

Section: Visualization and HALO software: why did the authors adjust the threshold for visualization? Will there be any subjective biases or errors that are introduced by this approach, rather than an unbiased automated approach?

Section: image processing with the Highplex FL module, I do not think that it is an accurate method to manually determine the positive threshold of each marker. Subjective method will always generate errors. There are some existing methods to determine the positive threshold:

[1] Rendeiro, André F., et al. "The spatial landscape of lung pathology during COVID-19 progression." *Nature* 593.7860 (2021): 564-569.

[2] Lu, Peng, et al. "IMC-Denoise: a content aware denoising pipeline to enhance Imaging Mass Cytometry." *Nature communications* 14.1 (2023): 1601.

Section: Image processing with the tissue classifier module and pixel classification: What is the specific ML algorithm that is being used in the HALO software? What are the training and predicting processes and data sets? This is a major component of layout out the methods and the capacity of other investigators to replicate this work.

Section: Single cell segmentation: What is the segmentation algorithm used for this task based on? : I wonder if this segmentation algorithm is developed from an existing pipeline:

[3] Zanotelli, Vito RT, and Bernd Bodenmiller. "ImcSegmentationPipeline: A pixelclassification based multiplexed image segmentation pipeline." Zenodo <https://doi.org/10.5281/zenodo.3841960> (2017).

The authors should cite it in the manuscript.

Furthermore, what kind of single cell information was extracted. Are these mean or median values?

Section: Immune cell classification and quantification of molecular expression: there are multiple imaging artifacts in IMC not just channel spillover. In most cases, the hot pixel removal should be considered (either through thresholding, fill in, or other methods). If the signal in a particular channel is low, then the authors may also consider denoising. What is the status of such processes in the data here?

Furthermore, why was the UMAP algorithm only used for 2 patients even though the dataset

is very large? why only 573 cells from each case were used? This is confusing.

Why did the authors apply Flow SOM algorithm with the UMAP results? UMAP is mainly used for visualization... in the dimensional reduction process there is also the potential for data to be lost. The authors are encouraged to cluster their raw data directly, and further there are some advanced methods for clustering such as Phenograph that are now considered in many contexts for these types of data to be superior. Finally, how were the parameters for FlowSOM selected here?

Minor figure question: in Figure 1, the authors use the acronym NE. Does this refer to not evaluated?

How many tumor regions were chosen based on CD20 imaging for the data in Figure 2? Were these selected manually? and what was the field of view of this region? In Figure 2, the percent fraction of cells that are illustrated, is this the percent fraction of TIL? or total cells?

Reviewer #2 (ESCC)(Remarks to the Author):

Authors report that CD39+ PD-1+ CD8+ T cells with an exhausted phenotype, identifying two distinct cell populations: precursor exhausted T (CD39+ Tpex) cells and differentiated exhausted T (CD39+ dTex) cells. Higher density of these cells, especially CD39+ Tpex cells in tumor tissue correlated with ICB benefit, which show that the unique subpopulation of CD39+ PD-1+ CD8+ T cells is crucial for ICB benefit, and suggest that these cells are key in TLS-mediated immune responses against tumors. As described below, there are several major and minor issues:

Major

1. The manuscript lacks logic, and each paragraph under the title lacks transitional sentences, such as "To characterize....., Given that....., To further investigate....." and so on.
2. The citation order of the figures in the manuscript is severely disordered and disorderly. Such as "Fig.S5E and Fig.S5D in page 7 line 8 and line 9, Fig.S6A-D in page 7 line 17- and line 19 and so on", too many citation order issues, please carefully check.
3. In the supplementary information: Fig. S7A, S8C, S8D, S11 and S12 are not described, why? Where is S10 and S12? I can not find them!!
4. Your research has only discovered a group of cells that are crucial for immunotherapy and has been validated, but there is little research on the molecular mechanisms. Can you increase the experiments to reveal how these cells work?
5. There is an absence of complementary experiments, either from alternative data sources or, more ideally, wet-lab validation (such as animal experiments).

Minor

1. In page 4 line 5, there is no content on the treatment of CTLA4 inhibitors in references 1-4. Please provide more comprehensive references.
2. In page 6 line 9, "to ICB therapy (n=27) (Fig. 1d)", but in your figure legends was "n=22" in page 38 line 12, why?
3. In page 8 line 2, what is the "Figs. 2g and 3a"?

Reviewer #3 (Systems immunology)(Remarks to the Author):

This is an interesting study on the effect of anti-PD1 immune checkpoint blockade (ICB) on different populations of exhausted T cell in esophageal squamous cell cancer, in particular focusing on CD39+ PD-1+ TCF-1+ CD8+ precursor exhausted T cells (Tpex) and CD39+ PD-1+ TCF-1- CD8+ differentiated exhausted T cells (dTex), looking at them in the tumor and its surrounding stroma, in neighboring tertiary lymphoid structures and in peripheral blood. Higher density of Tpex cells in tumor tissue correlated with ICB benefit and they were found at high levels in tertiary lymphoid structures. Importantly, the level of both dTex and Tpex cells detected in peripheral blood increases following ICB, suggesting that they might play a role in the response to ICB.

Overall this is a good study which generates some interesting and novel hypotheses regarding the role of exhausted T cell populations in the response to immunotherapy.

With regard to the analysis of peripheral blood shown in Figure 5 and 6, the FACS analysis leads to the conclusion that there is a preferential increase of CD39+ Tpex cells in the blood of responders following ICB therapy. However, in the more detailed analysis by mass cytometry carried out on six patients before and after ICB, the increase in Tpex cells was not significant. Were these six patients responders or non-responders? The increase in Tpex (C4) that is seen in fig 5c appears to be largely driven by one patient. It would also be helpful to see the cell numbers (%PD-1+ CD8+) for each patient in the dTex C1 cluster, as is done for C4.

The imaging mass cytometry analyses carried out in fig 6d-f on pre and post treatment samples from the same three patients show a huge amount of variation. It is questionable whether a study on so few samples can provide meaningful information about the nature of the ICB response, especially given the very different nature of the second biopsy samples – metastases at two different body sites and a second sampling of the primary site.

At several points in the manuscript there is tendency to state that changes in Tex populations are causing clinical responses to therapy (e.g. subheading on p12). The study is observational and cannot make claims of this nature, only that these changes are associated with clinical response: causation cannot be proved from the existing data, so more cautious wording is needed.

Minor points:

Figure 1. Some acronyms are undefined in the legend or elsewhere: SLO, NE, NA

Figure 2. Title: these data do not show that the CD39+ PD-1+ TCF-1+ CD8+ T cells improve clinical outcome of anti-PD-1 therapy, but rather that they are associated with improved outcome.

Authors Response to Initial Comments:

General Remarks:

First, we would like to thank the reviewers for their careful reading and helpful comments. We have carefully considered all the feedback and conducted the necessary additional analyses.

Major changes include denoising the IMC data to improve the downstream analysis, based on the feedback provided by Reviewer #1. Consequently, in the revised manuscript, the figures related to IMC analysis, including existing Figs. 2b, 2c, 2d, 2e, 2f, 2h, 2i, all of Figure 3, all of Figure 4, Fig. 5h, and Figs. 6e, 6f, all of Figure S1, all of Figure S2, all of Figure S7, and Figure S12 A, C, D, have been updated to reflect the results of the reanalysis after denoising. Minor numerical changes were observed as a result of the reanalysis, but these minor changes have not been explicitly discussed in the response letter.

Additionally, in this revision, we have newly added all of Figure S3, Figure S4D, Figure S6G, all of Figure S11, and Figure S12E. Notably, Figures S11 and S12E are related to IMC denoising, and Figure S3 presents the results of the MC dataset analysis to reproduce and reinforce our findings. This MC dataset analysis was performed based on the feedback from Reviewer #2.

In the revised manuscript, the sections highlighted in yellow indicate the changes made in response to the reviewers' comments. Additionally, we added further explanations and corrected grammatical errors in the text and figures that were overlooked in the initial draft. We also included supplementary data, such as the FACS gating strategy in Figure S6G, which were initially overlooked, to provide a detailed explanation of FACS analysis.

Our detailed responses to the comments of the reviewers are as follows.

※Please note that in response to the comments, the order of the supplementary figures has been significantly changed to fit the necessary structure of the text. Please refer to the correspondence table below for details.

Before revision	After revision
Fig.S1	Fig.S10
Fig.S2	Fig.S12
Fig.S3	Fig.S9
Fig.S4	Fig.S8
Fig.S5	Fig.S1
Fig.S6	Fig.S2
Fig.S7	Fig.S7
Fig.S8	Fig.S4
Fig.S9	Fig.S5
Fig.S10	Fig.S6

Reviewer #1 (IMC) (Remarks to the Author):

The authors present an interesting and in depth study of the spatial distribution of immune subsets in esophageal cancer, with a focus on the exhausted T-cell subsets. Overall, the manuscript is impactful and readable. This reviewer has some issues that could be addressed by the authors that focus on methodological presentation and details, as well as on some of the results.

Response

Thank you for evaluating our research. To further demonstrate the robustness of methodology, we have closely discussed during this revision, particularly addressing the issues related to IMC, such as noise. Below, we have listed the specific measures we have taken point by point for your review:

Overview comment: the description the computational and imaging methods in the main manuscript is very short. The methods should be described in detail in the main manuscript rather than the supplementary materials as the approach here is still somewhat novel (leveraging IMC).

Response

Thank you for your valuable advice. We agree that it is crucial to describe the methods in detail to demonstrate reproducibility and data robustness. We also believe that all methods, not just those involving IMC but also those using other tools, should be thoroughly explained. Therefore, we have detailed all methods in the main manuscript. The Method section related to IMC has been summarized from page 21, line 1 to page 36, line 22 so we hope you will refer to it.

Section: Visualization and HALO software: why did the authors adjust the threshold for visualization? Will there be any subjective biases or errors that are introduced by this approach, rather than an unbiased automated approach?

Response

Thank you for your valuable advice regarding the HALO analysis. We agree that adjusting the minimum threshold for visualization to avoid capturing shot noise could introduce subjective biases. To address this concern, we have implemented the following two measures:

- 1) Noise suppression for markers involved in immune cell analysis.
- 2) Review of the visualizations by a pathologist blinded to clinical outcomes.

For point 1), we employed the IMC-denoise package to suppress shot noise and hot pixels. The use of this package, by improving visual clarity, enhances the accuracy of the manual identification of cells [1]. The image below (Figure S11B) confirms that IMC-denoising is as effective as fluorescence immunostaining (Figure S11C), even in low signal-to-noise channels like PD-1 and CD39.

Consequently, this package minimizes the impact of shot noise, making the adjustment of the minimum threshold to avoid capturing shot noise almost unnecessary.

For detailed procedures of IMC-denoising, please refer to page25, line 21 to page 26, line 16, and for the representative images generated by denoising, please refer to supplementary Figure S11. Consequently, we reanalyzed all IMC data presented in this study.

For point 2), we aimed to reduce subjectivity by having a pathologist, blinded to clinical outcomes, adjust the thresholds for visualization. Specifically, for markers critical to the conclusions of this study, the pathologist confirmed the staining quality of the denoised images through comparison with fluorescence immunostaining (page 36, lines 12-22) (Fig. S11C). The revised visualization process is detailed on page 29, lines 20-23 (method section), and we hope you will refer to it.

References

[1] Lu, Peng, et al. "IMC-Denoise: a content aware denoising pipeline to enhance Imaging Mass Cytometry." *Nature communications* 14.1 (2023): 1601.

page25, line 21 to page 26, line 16 (method section)

IMC image denoising

We noted the presence of hot pixels and shot noise in the IMC images obtained for several markers, which could potentially affect downstream analysis. To clearly visualize protein expression, particularly in channels with low signal-to-noise ratios, set precise positive thresholds, and improve cell clustering, we utilized the IMC-Denoise software package (51). This software was used to

analyze markers including CD3, CD4, CD8, CD20, PD-1, CD39, TOX, Ki67, CD45RA, CD45RO, HLA-DR, CD11c, FoxP3, TCF-1, LAG-3, OX-40, TIM-3, CD69, and CD25.

IMC-Denoise employs two main steps: a differential intensity map-based restoration (DIMR) for hot pixel removal and a self-supervised deep learning algorithm for shot noise image filtering (DeepSNiF) for shot noise suppression. DeepSNiF requires model training prior to implementation. We trained a separate model for each marker, resulting in the creation of a separate dataset for each marker from our IMC images after DIMR hot pixel removal. Each dataset comprised approximately 30,000 64x64 patches after data augmentation. Subsequently, all models were trained on a single NVIDIA RTX 3000 GPU over 200 epochs, using the default parameter settings of the software. After training, the DIMR-processed IMC images were denoised by DeepSNiF using their corresponding trained models. The denoised IMC images were then evaluated by a pathologist (T.I.) (Fig. S11A). The quality of these images was confirmed to be comparable in quality to immunofluorescence staining for CD39, PD-1, and CD8 (Fig. S11B, S11C).

page 29, lines 20-23 (method section)

These images were visualized by manually adjusting the threshold value of each marker's expression, followed by a review and additional adjustments by the pathologist (T.I.), who was blinded to the clinical outcomes, to refine cell phenotyping (Fig. S12B).

page 36, lines 12-22 (method section)

Immunofluorescence Staining

Fluorescence multiplex immunostaining was performed on FFPE samples from an enrolled patient to assess the staining quality of denoised images. Antigen retrieval was performed using Antigen Retrieval Solution pH 9 (Nichirei, Tokyo, Japan) and microwaving for 20 min. The primary antibodies used were as follows: PD-1 (Clone NAT105, ab52587, Abcam), CD39 (Clone ERP20627, ab223842, Abcam), and CD8 (Clone 4B11, PA0183, Leica Biosystems). All primary antibodies were incubated at RT for 90 min. The following secondary antibodies were used: anti-mouse IgG2b CF Dye 647 (Biotium), anti-mouse IgG1 CF Dye 488 (Biotium), and anti-rabbit IgG CF Dye 568 (Biotium). All secondary antibodies were incubated at RT for 40 min. DAPI was used as the nuclear counterstain.

Section: image processing with the Highplex FL module, I do not think that it is an accurate method to manually determine the positive threshold of each marker. Subjective method will always generate errors. There are some existing methods to determine the positive threshold:

[1] Rendeiro, André F., et al. "The spatial landscape of lung pathology during COVID-19 progression." *Nature* 593.7860 (2021): 564-569.

[2] Lu, Peng, et al. "IMC-Denoise: a content aware denoising pipeline to enhance Imaging Mass Cytometry." *Nature communications* 14.1 (2023): 1601.

Response

Thank you very much for your important comments regarding the threshold. We agree that it is essential to ensure objectivity in determining the threshold of markers. To enhance manual annotation for IMC images,

- 1) we applied the IMC-denoise package to improve visual clarity and reduce analysis errors.
- 2) Furthermore, for threshold setting of markers, we ensured professional and objective judgment by having the data reviewed by a pathologist blinded to clinical outcomes, referring to previous reports [1].

Given the limitations of noise in IMC, it seems that various appropriate methods ensuring objectivity are being examined and devised. For example, as you indicated, the papers you provided [2-3] use the Gaussian Mixture Model for threshold setting, employing a statistical clustering approach without directly observing the staining. This method ensures objectivity. On the other hand, there are other methods that set thresholds in advance [4], use ilastik's random forest algorithm [5], or adjust the quantile threshold to determine the most appropriate threshold for the staining [6]. However, in our actual analysis, we have already confirmed that the accuracy of unified threshold settings is often susceptible to the influence of batch effects between IMC samples. Considering this, the advantage of our method could be the ability to visually capture subtle positive cells and address batch effects. Furthermore, given the benefits of performing localization analysis and further spatial analysis based on tissue classification in HALO, we chose to perform denoising to optimize for manual annotation.

However, while IMC-denoise methods can reduce analysis errors caused by noise, HALO still involves subjective issues related to threshold setting. To address these concerns, we utilized the review of a pathologist. Furthermore, we confirmed that the finalized threshold setting by HALO shows a strong correlation ($R=0.827$, $p < 0.0001$) with the CD39⁺PD-1⁺CD8⁺ population identified by automated gating in detecting low signal-to-noise channels such as CD39 and PD-1 (Fig. S12E). The methods are described in detail on page 30, Lines 11-20. We appreciate your attention to this detail and would be grateful if you could review it.

References

- [1] Miller, B.C., Sen, D.R., Al Abosy, R. et al. Subsets of exhausted CD8⁺ T cells differentially mediate tumor control and respond to checkpoint blockade. *Nat Immunol* 20, 326–336 (2019).
- [2] Rendeiro, André F., et al. The spatial landscape of lung pathology during COVID-19 progression. *Nature* 593.7860 (2021): 564-569.

- [3] Lu, Peng, et al. IMC-Denoise: a content aware denoising pipeline to enhance Imaging Mass Cytometry. *Nature communications* 14.1 (2023): 1601.
- [4] Tietscher, S., Wagner, J., Anzeneder, T. et al. A comprehensive single-cell map of T cell exhaustion-associated immune environments in human breast cancer. *Nat Commun* 14, 98 (2023).
- [5] Moldoveanu D, et al. Spatially mapping the immune landscape of melanoma using imaging mass cytometry. *Sci Immunol.* 2022 Apr;7(70):eabi5072.
- [6] Wang XQ, et al. Spatial predictors of immunotherapy response in triple-negative breast cancer. *Nature.* 2023 Sep;621(7980):868-876.

page 30, Lines 11-20 (method section)

The thresholds for positivity of each marker's expression were manually determined for cell phenotyping, with adjustments made in increments of 0.1 to 0.2 to obtain the optimal threshold. These thresholds were finalized following review by a pathologist (T.I.), who based decisions on the individual staining characteristics of nuclei or membranes, thereby confirming cell phenotypes defined by the distinctive characteristics of each marker's expression (Fig. S12B). Specifically, for threshold setting in low signal-to-noise channels such as CD39 and PD-1, there was a strong correlation ($R=0.827$, $p<0.0001$) between the cell density of CD39⁺ PD-1⁺ CD8⁺ T cells identified through automated gating in Image processing A and the visual threshold setting by HALO, confirming sensible cut-off selection (Fig. S12E).

Section: Image processing with the tissue classifier module and pixel classification: What is the specific ML algorithm that is being used in the HALO software? What are the training and predicting processes and data sets? This is a major component of layout out the methods and the capacity of other investigators to replicate this work.

Response

Thank you very much for your valuable comments. Explaining the analysis methods in detail is crucial for ensuring reproducibility. The pixel classification for tissue classification was performed using the Random Forest algorithm within the Tissue Classifier Module. To address batch effects, the training and prediction processes within this module were conducted separately for each ROI. Given that the number of ROIs to be processed is not vast, processing them individually helped ensure accuracy. We have provided these details in the section "Image processing with the Tissue Classifier module" on page 31, lines 2-7. We would appreciate it if you could refer to this section.

page 31, lines 2-7 (method section)

Image processing with the Tissue Classifier module

Subsequent to cell phenotyping, tissue classification across the entire ROI was performed using the Random Forest algorithm within the Tissue Classifier module implemented in HALO. Within the tumor area, five types were classified: tumor parenchyma, tumor stroma, portions of the normal region, TLSs, and blank regions. The training and prediction steps in this module were performed individually for each ROI.

Section: Single cell segmentation: What is the segmentation algorithm used for this task based on? : I wonder if this segmentation algorithm is developed from an existing pipeline:

[3] Zanutelli, Vito RT, and Bernd Bodenmiller. "ImcSegmentationPipeline: A pixelclassification based multiplexed image segmentation pipeline." Zenodo <https://doi.org/10.5281/zenodo.3841960> (2017).

The authors should cite it in the manuscript.

Furthermore, what kind of single cell information was extracted. Are these mean or median values?

Response

Thank you for your valuable comment. It is indeed necessary to cite the relevant work, and we have included the citation on page 25, lines 15-17. Regarding your second point, when extracting single cell information from histoCAT, the mean pixel values for each marker were used. This information has been added on page 27, lines 18-20. We appreciate your attention to this detail and would be grateful if you could review it.

page 25, Lines 15-17 (method section)

"IMC image processing A", based on a previously reported algorithm (50), was utilized to generate spatial information for CD8+ T cell clustering within tumors.

page 27, lines 18-20 (method section)

The quality of cell segmentation was verified using histoCAT (ver. 1.761) (54), and single-cell measurements were extracted from all available channels using the mean pixel values for each segmented cell.

Section: Immune cell classification and quantification of molecular expression: there are multiple imaging artifacts in IMC not just channel spillover. In most cases, the hot pixel removal should be considered (either through thresholding, fill in, or other methods). If the signal in a particular channel is low, then the authors may also consider denoising. What is the status of such processes in the data here?

Response

Thank you very much for your important advice. We agree that shot noise and hot pixels are critical drawbacks in IMC applications. Therefore, we considered the use of the IMC-denoise package [1] an essential step in conducting this study. We appreciate your recommendation for denoising, which has indeed improved the reliability of our data. As mentioned above, the denoising methods are detailed on page 25, line 21 to page 26, line 16. We hope you find this clarification helpful.

Reference

[1] Lu, Peng, et al. "IMC-Denoise: a content aware denoising pipeline to enhance Imaging Mass Cytometry." *Nature communications* 14.1 (2023): 1601.

page 25, line 21 to page 26, line 16

Please refer to the above

Furthermore, why was the UMAP algorithm only used for 2 patients even though the dataset is very large? why only 573 cells from each case were used? This is confusing.

Response

Thank you very much for your valuable comments regarding the UMAP and clustering algorithm. Your feedback has been helpful in improving the scientific validity of our analysis. Following your suggestion, we have reanalyzed the data (Figs. 2d, S2).

To address your points, we performed clustering analysis using the FlowSOM algorithm and visualized these clusters with UMAP plots from normalized data. While we believe that Phenograph could be effectively used for this type of data, we chose the FlowSOM algorithm based on previously reported studies [1,2] that also demonstrate its suitability for imaging applications.

Specifically, to achieve accurate phenotyping of exhausted T cells within tumors, we considered it necessary to have a sufficient number of CD8⁺ T cells. 11 ROIs, each containing more than 500 CD8⁺ T cells, were automatically extracted for clustering analysis using FlowSOM. To ensure high-quality analysis, sampling was conducted with equal numbers, resulting in 555 CD8⁺ T cells being randomly used from each sample. UMAP analysis was then employed to visualize these clusters.

The methods section detailing this process can be found on page 28, lines 3-4 and lines 11-20. These reanalysis data are reflected in Figure S2, with some data also shown in Figure 2d. The corresponding analysis in the main text can be found on page 6, line 23 to page 7, line 2, and page 7, line 16 to page 8, line 12. We hope this clarifies our approach.

References

- [1] Hunter B, et al. OPTIMAL: An OPTimized Imaging Mass cytometry AnaLysis framework for benchmarking segmentation and data exploration. *Cytometry A*. 2024 Jan;105(1):36-53.
- [2] Rahim MK, et al. Dynamic CD8⁺ T cell responses to cancer immunotherapy in human regional lymph nodes are disrupted in metastatic lymph nodes. *Cell*. 2023 Mar 16;186(6):1127-1143.e18.

page 6, line 23 to page 7, line 2 (result section)

UMAP plots of representative CD8⁺ T cells revealed a distinct subset of PD-1hi cells co-expressing TOX, Ki67, and CD39 (Fig. 2d), which are associated with T-cell exhaustion (6, 8), proliferative activity, and tumor specificity (11-16), respectively.

page 7, line 16 to page 8, line 12 (result section)

CD39⁺ T_{pex} cells identified by TCF-1 expression constitute a distinct group among CD39⁺ PD-1⁺ T_{ex} cells

We hypothesized that the phenotypes of CD8⁺ T cells might differ within the cohort of CD39⁺ PD-1⁺ T_{ex} cells. From 11 ROIs, each containing more than 500 CD8⁺ T cells, we applied unsupervised clustering to CD8⁺ T cells using FlowSOM (Fig. S2A) and visualized these clusters in UMAP plots (Fig. S2B, S2C). We identified five CD39hi PD-1hi T_{ex} clusters (C1–5) characterized by high TOX and PD-1 expression (Fig. S2A). When mapping these clusters to the representative ROIs of tumors lacking TLSs (Fig. S2D) and tumors containing TLSs (Fig. S2E), clusters C1 and C2, which exhibited high levels of TCF-1 expression and correspond to T_{pex} cells due to their exhaustion profile, were primarily localized to the stroma and TLSs. The localization characteristics of these TCF-1⁺ populations to TLSs were consistent with previously reported findings (29, 30). In contrast, clusters C3–5, primarily localized within the tumor parenchyma, corresponded to differentiated T_{ex} (dT_{ex}) cells lacking TCF-1 expression (Fig. S2D, S2E). The presence of CD39⁺ T_{pex} (TCF-1⁺ CD39⁺ PD-1⁺ CD8⁺ T) cells, previously reported in several studies (26, 32, 33), was also confirmed through co-staining in the tumor ROIs (Figs. 2g, S11B). To characterize this unique population within the TME, we categorized CD39⁺ PD-1⁺ T_{ex} cells from 27 cases as CD39⁺ T_{pex} and CD39⁺ dT_{ex} (TCF-1⁻ CD39⁺ PD-1⁺ CD8⁺ T) cells. Both subsets exhibited high exhaustion marker levels (Fig. S1A, S1B), suggesting that they are potentially tumor-reactive T cells under continuous antigen stimulation in the TME (6-9).

page 28, lines 3-4 (method section)

Subsequently, cell data were normalized to the 99th percentile and then subjected to Z-score normalization (51, 56)

page 28, lines 11-20 (method section)

High-dimensional data analysis of IMC tumor data

FlowSOM and UMAP analyses were performed using Cytobank (<https://cytobank.org>). To ensure a sufficient number of cells for Tex phenotype verification, FlowSOM analysis (32, 56) was performed using equal sampling from each of the 11 ROIs, each containing over 500 CD8⁺ T cells. The analysis identified 225 clusters and 7 metaclusters specifically for CD8⁺ T cells. Key markers used to identify the Tex clusters included OX40, LAG3, TIM3, TCF1, TOX, PD-1, CD45RA, CD69, Ki67, CD39, and CD25 with an arcsinh cofactor of 1 for parameter transformation. For visualizing the phenotype, the resulting clusters were analyzed using UMAP, applying the same parameters as in FlowSOM, with settings adjusted to 15 neighbors and a minimum distance of 0.01.

Why did the authors apply Flow SOM algorithm with the UMAP results? UMAP is mainly used for visualization... in the dimensional reduction process there is also the potential for data to be lost. The authors are encouraged to cluster their raw data directly, and further there are some advanced methods for clustering such as Phenograph that are now considered in many contexts for these types of data to be superior. Finally, how were the parameters for FlowSOM selected here?

Response

Thank you very much for your important comments. As mentioned in the previous section, we reanalyzed the data using the methods described there. All relevant parameters were selected for phenotyping exhausted T cells as described in page 28, lines 16-18.

Furthermore, we also reanalyzed the IMC data for lymph nodes using the same clustering approach to phenotype exhausted T cells (Fig. S7) as we judged that it would enhance the reliability of the data (we did not perform clustering analysis for SLOs previously). The panels used for the lymph nodes differed from those for the tumor ROIs, so clustering analyses were performed separately. Although the conclusions remain unchanged from the pre-revision analysis, the results are presented in Figure S7. The methods are described on page 28, Line 22 to page 29, line 7. The corresponding analysis in the main text can be found on page 14, line 12 to page 15, line 16.

We apologize for any inconvenience this may cause and appreciate your review of these additional analyses.

page 28, line 16-18 (method section)

Key markers used to identify the Tex clusters included OX40, LAG3, TIM3, TCF1, TOX, PD-1,

CD45RA, CD69, Ki67, CD39, and CD25 with an arcsinh cofactor of 1 for parameter transformation.

page 28, Line 22 to page 29, line 7 (method section)

High-dimensional data analysis of IMC SLO data

CD8⁺ T cells extracted from SLOs were analyzed separately due to the distinct panels used for tumors. All cells from three samples were included in the downstream analysis, reflecting the abundance of T cells. For each SOM, 225 clusters and 10 metaclusters were identified for CD8⁺ T cells. The following markers were used to identify the Tex clusters: OX40, LAG3, TIM3, TCF1, TOX, PD-1, CD45RA, CD69, Ki67, CD39, CD25, CD45RO, and HLA-DR with an arcsinh cofactor of 1 for parameter transformation. For phenotype visualization, the resulting clusters were analyzed using UMAP, employing the same parameters as in FlowSOM, with the algorithm settings adjusted to 15 neighbors and a minimum distance of 0.01.

page 14, line 12 to page 15, line 16 (result section)

CD39⁺ T_{pex} cells in secondary lymphoid organs (SLOs) exhibit an exhausted profile shared with counterpart cells in the blood and tumor

Circulating tumor-specific T_{pex} cells are reportedly recruited from lymph nodes (LNs) into the TME (32, 44, 45). Consequently, we hypothesized that CD8⁺ T cells with a CD39⁺ PD-1⁺ Tex phenotype are present within SLOs. Therefore, we performed unsupervised clustering analysis of CD8⁺ T cells manually extracted from the ROIs of individually resected SLOs from three patients (Fig. S7B) and visualized these clusters using UMAP analysis (Fig. S7C). We identified clusters of CD39^{hi} PD-1^{hi} Tex cells (C1–5) with high levels of TOX expression. The presence of this CD39⁺ PD-1⁺ Tex subset was also confirmed in a representative SLO image (Fig. 5h). Although only a single case of HNSCC in the ICB cohort had non-metastatic LN samples available post-ICB therapy for MC analysis, we found a remarkably abundant population of CD39⁺ PD-1⁺ and CD39⁺ T_{pex} cells within the CD8⁺ T cells (Fig. S3I). Among these clusters, C5 exhibited relatively low TCF-1 expression, possibly reflecting CD39⁺ dTex cells. However, most importantly, C1 and C2 which exhibited high levels of Ki67 expression also expressed high levels of TCF-1 (Fig. S7B), corresponding to CD39⁺ T_{pex} cells. This characteristic of high Ki67 expression was similar to the findings in the dataset of resected LNs from the HNSCC that did not receive ICB treatment, where CD39⁺ T_{pex} cells had higher Ki67 expression rates than CD39⁺ dTex cells (Fig. S3J).

To elucidate the spatial characteristics of these clusters within LNs, we mapped each CD8⁺ T cell cluster as shown in Figure. S7A and S7D, and analyzed their proximity to major immune cells manually extracted from each ROI. We found that all CD8⁺ T cell clusters were generally in close proximity to CD4⁺ T cell subsets (Fig.S7E), compared to CD20⁺ B cells and antigen-presenting cells (CD11c⁺ HLA-DR⁺). Notably in the representative SLO, CD39⁺ T_{pex} clusters (C1-4) were

localized in significantly closer proximity to PD-1+ Treg cells compared to other non-exhausted CD8+ T cells (Fig. S7F). This proximity characteristic was consistently observed in all three cases (Fig. S7G) and suggests possible immunosuppressive conditions surrounding CD39+ PD-1+ Treg cells in LNs (32, 44, 45).

Minor figure question: in Figure 1, the authors use the acronym NE. Does this refer to not evaluated?

Response

Thank you for your comment. The acronym "NE" in Figure 1 refers to "not evaluable." This has been noted in the figure legend of Figure 1. Please refer to page 55, lines 6-8 for more details.

page 55, lines 6-8 (figure legend)

(b) Clinical data and samples analyzed for each of the 31 ESCC patients. R: responder, NR; non-responder, NA: not applicable, NE: not evaluable.

How many tumor regions were chosen based on CD20 imaging for the data in Figure 2? Were these selected manually? and what was the field of view of this region?

Response

Thank you for your questions. In response to your feedback, we have revised the criteria for ROI selection and detailed these in the Methods section on page 24, lines 6-22.

For ROI acquisition, one tumor region was randomly selected based on the CD3 IHC image per sample, and one TLS region in tumor-adjacent or -inherent areas was also randomly selected based on both the CD3 and CD20 IHC images per sample. The representative images shown in Figure 2a include multiple TLSs analyzed; these data were obtained to test the feasibility of ROI selection based on IHC images. We have also noted that the area of the acquired regions ranges from 0.39 to 2.25 mm².

These details have been thoroughly described, and we hope you will refer to them.

page 24, lines 6-22 (method section)

Selection of ROIs for IMC analysis

Whole immunohistochemical-stained slides from all cases were scanned to identify areas rich in CD3+ cells and CD3+/CD20+ TLSs within the tumor. For tumors lacking TLS, areas with a high density of CD3+ cells were randomly mapped from consecutive IHC sections stained for CD3. One ROI per mapped area was then acquired from IMC-stained samples (Fig. S8A). Before the

acquisition of ROIs within TLSs, TLSs were initially screened in tumor-adjacent or -inherent areas using consecutive IHC sections stained for CD3 and CD20. Subsequently, randomly selected TLSs were then mapped to their corresponding locations on IMC-stained samples, ensuring that at least one tumor site per TLS was included. The accuracy of acquiring designated ROIs was experimentally confirmed twice using the representative sample (patient ID KU37) (Fig. 2a). Upon verification, for each sample in which TLSs were detectable, ROIs corresponding to TLSs and associated tumor areas were captured (Fig. S8C). Optionally, additional tumor ROI without TLSs, previously mapped based on IHC-stained serial sections, were also acquired. Given the predefined conditions described above, the area for ROI acquisition was predetermined, and ROIs were acquired within a range of up to 2.25 mm² (0.39–2.25 mm²).

In Figure 2, the percent fraction of cells that are illustrated, is this the percent fraction of TIL? or total cells?

Response

Thank you for your question. The percent fraction shown in Figure 2 represents the total cells. We have made corrections to Figure 2b and Figure 2c (page 56) and updated the corresponding figure legend (page 57, lines 8-11). We hope you will refer to these changes.

page 57, lines 8-11 (figure legend)

Each fraction (%) of the annotated immune profile among total cells from the tumor regions of interest (ROIs) for each of the 27 enrolled patients at pre-treatment (bottom). (c) Proportion (%) of PD-1⁺ CD8⁺ T cells among total cells from the tumor ROIs between Rs (n=9) and NRs (n=13) (n=22); Mann-Whitney U-test.

We look forward to hearing from you regarding our submission. We would be glad to respond to any further questions and comments that you may have.

Reviewer #2 (ESCC) (Remarks to the Author):

Authors report that CD39+ PD-1+ CD8+ T cells with an exhausted phenotype, identifying two distinct cell populations: precursor exhausted T (CD39+ Tpex) cells and differentiated exhausted T (CD39+ dTex) cells. Higher density of these cells, especially CD39+ Tpex cells in tumor tissue correlated with ICB benefit, which show that the unique subpopulation of CD39+ PD-1+ CD8+ T cells is crucial for ICB benefit, and suggest that these cells are key in TLS-mediated immune responses against tumors. As described below, there are several major and minor issues:

Response

Thank you for your evaluation of our research. The points you have raised are extremely helpful in improving our study, and we believe that adding data to support our hypothesis has enhanced the value of our work. Please see the responses below.

Major

- 1. The manuscript lacks logic, and each paragraph under the title lacks transitional sentences, such as “To characteriaed....., Given that....., To further investigate.....”and so on.**

Response

We truly appreciate your guidance in improving the clarity and logic of our manuscript. In response to your advice, we have made the necessary revisions to enhance the logical flow of the text and clarify our findings. Specifically, we have updated the following sections:

Page 6, line 10: Added "Consequently" to better transition between ideas.

Page 6, line 15: Inserted "To further investigate the major determinants of ICB responses" to explicitly state the purpose of our analysis.

Page 9, line 6: Prefaced with "Given these findings" to connect the results with subsequent discussion points.

Page 9 lines 22-24: We clarified the objective by starting with "To explore the interplay...!"

Page 10, line 23: Included "To explore the relationship with ICB therapy" to clarify the focus of our study on therapeutic interactions.

Furthermore, we have ensured that the explanation of new data throughout the manuscript is consistently logical and well-integrated, providing a clearer understanding of our research findings.

The citation order of the figures in the manuscript is severely disordered and disorderly. Such as “Fig.S5E and Fig.S5D in page 7 line 8 and line 9, Fig.S6A-D in page 7 line 17- and line 19 and so on”, too many citation order issues, please carefully check.

Response

We apologize for the confusion caused by the initial arrangement of figures in our manuscript. We appreciate your point regarding the unclear sequence, which highlighted an important aspect of our presentation. Following your suggestion, we have thoroughly revised the order of the figures to ensure that they align more logically with the text and facilitate a smoother understanding for the readers. We have also added new Supplementary Figures, which have been renumbered to reflect these changes effectively. Please refer to the updated sequence directly in the revised manuscript, where the new figures and their order are designed to enhance the narrative flow and aid in the better comprehension of our study’s findings.

2. In the supplementary information: Fig. S7A, S8C, S8D, S11 and S12 are not described, why? Where is S10 and S12? I can not find them!!

Response

We sincerely apologize for the oversight in the ordering and referencing of figures within the supplementary information of our manuscript. We have conducted a thorough review and corrected the figure numbering to ensure all figures are accurately described. Please refer to the revised supplementary information directly, where the updated figures and their descriptions should now clearly support the data presented.

4. Your research has only discovered a group of cells that are crucial for immunotherapy and has been validated, but there is little research on the molecular mechanisms. Can you increase the experiments to reveal how these cells work?

Response

Thank you very much for your valuable comments and the opportunity to address the concerns raised regarding our manuscript.

We appreciate your pointing out the crucial need to delve deeper into the molecular mechanisms of CD39⁺ TpeX cells within the biological context of lymph nodes and tumors. Indeed, your insight highlights important areas such as the behavioral dynamics of these cells, the timing of CD39 expression, specific differences from normal TpeX cells, and their role within the broader spectrum

of exhausted cells, which merit further investigation.

We fully agree with the necessity of further verification of these cell behaviors as you suggested, particularly from the perspective of molecular mechanisms. However, a major issue is that we have almost depleted the existing live patient samples necessary for this research. In a separate project, we plan to elucidate the molecular mechanisms of T cell exhaustion within lymphoid tissues, and we believe it is prudent to use the findings of our current research as a stepping stone for subsequent investigations.

In light of this, we have carefully reviewed and revised our discussion of the limitations and future directions of our research to reflect these considerations more accurately (please see revised sections on page 19 line 22 to page 20 line 10). Furthermore, as described in the next comment, we have used the dataset to substantiate our findings and demonstrate that CD39⁺ Tpex is a unique population.

We hope our revisions and responses address your concerns satisfactorily.

page 19 line 22 to page 20 line 10 (discussion section)

Limitations of our study include the absence of fresh and post-treatment tumor samples, which prevented functional assessment of the tumor specificity of CD39⁺ Tpex cells. Additionally, we could not detect all tumor-specific Tpex cells, as these cells often exhibit low CD39 expression early in exhaustion (47). Variability in CD39 expression in Tpex cells has been reported (26, 32, 33, 48), and our evidence regarding the tumor specificity of CD39⁻ TCF-1⁺ PD-1⁺ CD8⁺ T cells was insufficient. Further research is needed on the TCR repertoire and tumor antigen reactivity of CD39⁺ Tpex cells in tumors and blood. Moreover, the IMC analysis post-ICB therapy was constrained by a small number of cases, late time points, and the inclusion of metastatic tissues. Given the limitations of this observational study, direct evidence regarding the changes in the TME triggered by ICB therapy remains lacking. Further verification is needed to understand how CD39⁺ Tpex cells are activated following ICB and how they contribute to antitumor activity within the TME.

5. There is an absence of complementary experiments, either from alternative data sources or, more ideally, wet-lab validation (such as animal experiments).

Response

Thank you for your important suggestion regarding the necessity of complementary studies to ensure the reproducibility of our research findings. We agree with its necessity and have conducted an analysis using the MC dataset [1] to further substantiate our hypothesis. In this analysis, we identified CD39⁺ Tpex cells within both tumors and lymph nodes. We selected this dataset for several reasons: (1) it is linked to a rigorous clinical trial involving patients treated with ICB, (2) the cancer type is squamous cell carcinoma, and (3) the analysis utilizes actual tumor samples and

lymph nodes. These analyses are presented in the newly created Figure S3.

Notably, our findings demonstrate that intratumoral CD39⁺ T_{pex} cells possess a unique profile characterized by high proliferative activity despite their exhausted phenotype (Figs. S3B, S3D), thereby confirming the reproducibility of our data in tumor, blood and SLO.

Although we cannot analyze the treatment effects of ICB therapy with this dataset due to the protocol of this clinical study [1], we found a significant positive correlation between the frequencies of CD39^{hi} PD-1^{hi} CD8 T cells and CD39⁺T_{pex} cells (Fig. S3E), which supports the results of our study (Fig. 2h). Moreover, the correlation suggests that CD39⁺ T_{pex} cells may serve as a biomarker for ICB efficacy, in line with the findings from our study (Figs.2i, S1D) and previous reports [2].

The relevant text can be found on pages 8 (lines13-24), pages 9 (lines 14-16), pages 14 (lines 21-24), pages 15 (lines 4-6), pages 16 (lines 12-21), page34 (lines11-21), and page 34 (line 23) to page 35 (line 4)

We hope this additional analysis will address your concerns.

References:

[1] Cell 186, 1127-1143.e18 (2023).

[2] Immunity 56, 1–14 (2023).

pages 8 (lines13-24) (result section)

To obtain further substantiation of their existence, we analyzed a dataset (32) from a cohort of head and neck squamous cell carcinoma (HNSCC) patients treated with atezolizumab (anti-PD-L1), where tumor-infiltrating lymphocytes (TILs) were examined using mass flow cytometry (MC) (Fig. S3A). We identified populations of CD39^{hi} PD-1^{hi} clusters (Clusters; C1, C2, C3, C4) through unsupervised clustering of CD8⁺ T cells with FlowSOM (Fig. S3B) and visualized them using UMAP plots (Fig.S3C). Notably, clusters C3 and C4, which correspond to CD39⁺ T_{pex} cells, were characterized not only by their expression of TCF-1, but also their relatively higher expression of CCR7, CD127, and TIGIT (Fig. S3B, S3D), indicating that they are phenotypically distinct T_{pex} cells, unlike CD39⁺ dT_{ex} cells (24, 26, 31, 32). These results collectively indicate that CD39⁺ PD-1⁺ CD8⁺ T cells constitute a heterogeneous group that includes not only dT_{ex} cells but also T_{pex} cells.

pages 9 (lines 14-16) (result section)

A similar correlation was also observed using the ICB cohort dataset (32) (n=8, r=0.831, P = 0.0106; Fig. S3E).

pages 14 (lines 21-24) (result section)

Although only a single case of HNSCC in the ICB cohort had non-metastatic LN samples available

post-ICB therapy for MC analysis, we found a remarkably abundant population of CD39⁺ PD-1⁺ and CD39⁺ Tpex cells within the CD8⁺ T cells (Fig. S3I).

pages 15 (lines 4-6) (result section)

This characteristic of high Ki67 expression was similar to the findings in the dataset of resected LNs from the HNSCC that did not receive ICB treatment, where CD39⁺ Tpex cells had higher Ki67 expression rates than CD39⁺ dTex cells (Fig. S3J).

pages 16 (lines 12-21) (result section)

However, CD39⁺ Tpex cells exhibited consistently high Ki67 expression in the second biopsy, a pattern also observed in our IMC results prior to ICB therapy (Fig. S3B). This result is further supported by the characteristic high Ki67 expression of intratumoral CD39⁺ Tpex cells in the HNSCC dataset, which includes 6 out of 8 samples post-ICB therapy (Figs. S3B, S3C, S3D). Manual gating also confirmed that CD39⁺ Tpex cells exhibited high Ki67 expression comparable to CD39⁺ dTex cells (Figs. S3F, S3G). Furthermore, TCF-1⁺ Ki67⁺ CD8⁺ TILs, a key predictor of ICB efficacy in breast cancer (38), showed a significantly higher CD39⁺ PD-1⁺ positivity rate compared to TCF-1⁺ Ki67⁻ CD8⁺ TILs in the HNSCC dataset (Fig. S3H), indicating that many highly proliferative TCF-1⁺ CD8⁺ T cells are reflective of CD39⁺ Tpex cells.

page34 (lines11-21) (method section)

High-dimensional data analysis of MC dataset

FlowSOM and UMAP analyses were performed on eight available tumor samples, obtained through biopsy or resection, from the ICB-treated cohort of HNSCC patients (32) using Cytobank. CD8⁺ T cells were manually gated, as described in Figure S3I. FlowSOM analysis employed equal sampling for consistency across samples. For each SOM, 225 clusters and 9 metaclusters were identified for CD8⁺ T cells. The markers used for this analysis included PD-1, CD39, TCF-1, Ki67, TIM-3, CD127, CD69, CD103, GranzymeB, Tbet, CD39, CD38, CD25, CD45RA, CCR7, CD27, TCF1, TIGIT, and HLA-DR. The resulting clusters were then analyzed using UMAP, employing the same parameters as in FlowSOM with the algorithm adjusted for 15 neighbors and a minimum distance of 0.01.

page 34 (line 23) to page 35 (line 4) (method section)

Cell quantification of MC dataset

Quantification by manual gating was performed on 8 tumor samples, 1 non-metastatic lymph node (LN) sample from the dataset of the ICB cohort of HNSCC patients, and 9 non-metastatic LN samples from the standard of care cohort (32). The expression rates of TCF-1, PD-1, CD39, and

Ki67 in CD8⁺ T cells were determined based on the gating strategy shown in Figure S3I.

Minor

- 1. In page 4 line 5, there is no content on the treatment of CTLA4 inhibitors in references 1-4. Please provide more comprehensive references.**

Response

Thank you very much for your valuable comment. We have added a recent review article as reference 5 to provide a more comprehensive citation. As you pointed out, mentioning CTLA4 inhibitors, in addition to PD-1 inhibitors, is necessary for the standard treatment of unresectable advanced esophageal cancer. We have made corrections to the text accordingly. Please refer to the revised text on Page 4, Lines 3-7.

Page 4, Lines 3-7 (introduction section)

Immunotherapy, particularly the use of immune checkpoint blockade (ICB) drugs to target PD-1 or CTLA4, has significantly improved the prognosis for esophageal cancer (1-5). PD-1 inhibitors are not only used as monotherapy; they can also be also combined with CTLA4 inhibitors and cytotoxic agents, establishing them as key components of systemic chemotherapy for esophageal cancer (1, 5).

- 2. In page 6 line 9, “to ICB therapy (n=27) (Fig. 1d)”, but in your figure legends was “n=22” in page 38 line 12, why?**

Response

Thank you for your important comment. The NE cases refer to those where the tumor response to ICB treatment could not be evaluated according to RECIST v1.1. These cases were excluded from the efficacy analysis, as noted in the Figure legend (page 55, lines 11-14). The reasons for NE are additionally provided in the Methods section (page 22 lines 12-16).

page 55, lines 11-14 (figure legend)

(d) Representative immunostaining of PD-L1 for an ESCC patient (left). CPS (combined positive score) in Rs (responders; n=9) and NRs (non-responders; n=13) (n=22) (right). Five NE cases were excluded from this analysis; Mann-Whitney U-test.

page 22 lines 12-16 (method section)

Cases not evaluable (NE) included 3 patients changing treatment, including radiotherapy prior to the initial treatment evaluation, and 2 patients dying before treatment evaluation could be conducted. NE

cases were excluded from the analysis based on treatment efficacy analysis.

3. In page 8 line 2, what is the “Figs. 2g and 3a”?

Response

Thank you for your important feedback. We apologize for any unclear expressions. To clarify, we have revised the text on page 8, lines 6-8, to better explain the visualization of channels and the detection of CD39⁺ Tpex cells. Please refer to Figures 2g and S11B for further details.

page 8, lines 6-8 (result section)

The presence of CD39⁺ Tpex (TCF-1⁺ CD39⁺ PD-1⁺ CD8⁺ T) cells, previously reported in several studies (26, 32, 33), was also confirmed through co-staining in the tumor ROIs (Figs. 2g, S11B).

We look forward to hearing from you regarding our submission. We would be glad to respond to any further questions and comments that you may have.

Reviewer #3 (Systems immunology)(Remarks to the Author):

This is an interesting study on the effect of anti-PD1 immune checkpoint blockade (ICB) on different populations of exhausted T cell in esophageal squamous cell cancer, in particular focusing on CD39+ PD-1+ TCF-1+ CD8+ precursor exhausted T cells (Tpex) and CD39+ PD-1+ TCF-1- CD8+ differentiated exhausted T cells (dTex), looking at them in the tumor and its surrounding stroma, in neighboring tertiary lymphoid structures and in peripheral blood. Higher density of Tpex cells in tumor tissue correlated with ICB benefit and they were found at high levels in tertiary lymphoid structures. Importantly, the level of both dTex and Tpex cells detected in peripheral blood increases following ICB, suggesting that they might play a role in the response to ICB.

Overall this is a good study which generates some interesting and novel hypotheses regarding the role of exhausted T cell populations in the response to immunotherapy.

Response

We appreciate your recognition of the novelty and relevance of our study. The findings presented in this research contribute to the growing body of knowledge on the immune response to ICB therapy in ESCC. By elucidating the roles of Tpex and dTex cells in various anatomical contexts, we aim to advance the development of more effective and personalized immunotherapeutic strategies. Regarding the following comments, we have carefully considered and addressed each one, and we kindly request your review and feedback on our responses.

With regard to the analysis of peripheral blood shown in Figure 5 and 6, the FACS analysis leads to the conclusion that there is a preferential increase of CD39+ Tpex cells in the blood of responders following ICB therapy. However, in the more detailed analysis by mass cytometry carried out on six patients before and after ICB, the increase in Tpex cells was not significant. Were these six patients responders or non-responders?

Response

Thank you for your important observation regarding the interpretation of the results from the patients analyzed by Mass Cytometry (MC). We agree that it is crucial to verify whether the changes observed in C4 detected by MC after ICB therapy align with our hypothesis in FACS, particularly considering whether the six patients were responders or non-responders. In response to your comment, we have created Figure S4D to illustrate the comparison between clustering and manual gating quantifications from each sample, and included the relevant text in page 13 lines 7-12.

The peripheral blood samples used for MC analysis included three responders and three non-

responders, each preserved for subsequent analysis. Regardless of treatment efficacy, the quantitative analyses by both FACS and MC showed an increase in the frequency of CD39⁺ Tpex (CD39+PD-1+Ki67+TCF-1+CD8⁺ cells) following ICB administration. This observation aligns with the rationale for the increase in cluster C4 in non-responders. Conversely, among the responders, one out of three cases showed no change in frequency post-ICB administration, which contrasts with the quantitative results from FACS and MC.

Upon reanalyzing the frequency comparisons between C4 and the manually quantified CD39⁺ Tpex from FACS and MC, we observed a strong correlation among the three measurements (Fig. S4D), indicating that they likely reflect the same cell population to a certain extent. Although there is a discrepancy in the frequencies between manual gating quantification and clustering-derived C4, we attribute this to the differences in analytical methods involving a rich set of parameters. Furthermore, one notable increase was observed in patient (ID KU37), as you specifically pointed out in the following section, consistent across both manual quantification by FACS and MC (Fig. S4D), indicating they reflect the almost same population. Therefore, the minor discrepancies with FACS are considered to be due to the differences in analytical parameters. However, since the purpose of the MC analysis was to elucidate the detailed phenotype of CD39⁺ Tpex cells, we believe that the minor differences in frequency changes post-ICB administration are within an acceptable range.

We hope this clarification addresses your concerns.

page 13 lines 7-12 (result section)

Despite the lower frequency of cluster C4 compared to the manually gated groups from both FC and MC, a strong correlation was observed between the frequency of C4 and each manually gated group (Fig. S4D). Notably, a significant increase in C4 was observed in one particular case (Patient ID KU37), with similar results also reported in manually gated groups (Figs.5c, S4D).

The increase in Tpex (C4) that is seen in fig 5c appears to be largely driven by one patient. It would also be helpful to see the cell numbers (%PD-1+ CD8+) for each patient in the dTex C1 cluster, as is done for C4.

Response

Thank you for highlighting this important point. In response to your points, we have also included the frequencies of the C1 (CD39⁺ dTex cluster) in Figure 5c.

As mentioned in the previous section, the increase in C4 observed in one patient (Patient ID KU37: responder) following treatment indeed contributed significantly to the overall increase in Tpex cells. This notable increase was consistently observed in different quantification analyses, both by FACS and MC manual gating. Additionally, in alignment with the findings reported in Cell 186,

1127-1143.e18 (2023), Tplex cells are present at very low frequencies in the blood, indicating that they are similar to the frequencies we are reporting. We hope that this additional information, along with the previous details provided, addresses your concerns and aids in your understanding of our findings.

The imaging mass cytometry analyses carried out in fig 6d-f on pre and post treatment samples from the same three patients show a huge amount of variation. It is questionable whether a study on so few samples can provide meaningful information about the nature of the ICB response, especially given the very different nature of the second biopsy samples – metastases at two different body sites and a second sampling of the primary site.

Response

Thank you for your insightful comments regarding the imaging mass cytometry analyses in Fig. 6d-f on pre- and post-treatment samples from the same three patients. We agree with your observation that the post-treatment samples include metastatic tumors, which makes direct comparison with pre-treatment data challenging. It is indeed important to carefully interpret the significance of the data from such samples.

In response to your comments, instead of comparing the results across these samples, we focused on the consistent high levels of Ki67 expression in CD39⁺ Tplex cells, as interpreted in Fig. 6f. We have made substantial revisions to the content from page 16, line 12 to page 17, line 2, and also significantly changed the content under the subheading on page 16, lines 2-3.

This revised content highlights that, based on our analysis results and those from the dataset, CD39⁺ Tplex cells exhibit high levels of Ki67 expression not only in tumor tissues but also in lymph nodes and blood. As referenced in the latest report [1], these findings support the hypothesis that CD39⁺ Tplex cells are associated with the antitumor effects mediated by ICB. We hope you find these changes satisfactory.

Furthermore, we acknowledge the limitations of our study due to the observational nature of the research, which restricts the extent of meaningful analysis of immune dynamics within the tumor microenvironment post-ICB treatment. We have added a statement addressing this limitation in our manuscript. These changes can be found on page 20, lines 5-10 in the revised manuscript.

references

[1] Wang XQ, et al. Spatial predictors of immunotherapy response in triple-negative breast cancer. *Nature*. 2023 Sep;621(7980):868-876.

page 16, line 12 to page 17, line 2 (result section)

However, CD39⁺ T_{pex} cells exhibited consistently high Ki67 expression in the second biopsy, a pattern also observed in our IMC results prior to ICB therapy (Fig. S3B). This result is further supported by the characteristic high Ki67 expression of intratumoral CD39⁺ T_{pex} cells in the HNSCC dataset, which includes 6 out of 8 samples post-ICB therapy (Figs. S3B, S3C, S3D). Manual gating also confirmed that CD39⁺ T_{pex} cells exhibited high Ki67 expression comparable to CD39⁺ dTex cells (Figs. S3F, S3G). Furthermore, TCF-1⁺ Ki67⁺ CD8⁺ TILs, a key predictor of ICB efficacy in breast cancer (38), showed a significantly higher CD39⁺ PD-1⁺ positivity rate compared to TCF-1⁺ Ki67⁻ CD8⁺ TILs in the HNSCC dataset (Fig. S3H), indicating that many highly proliferative TCF-1⁺ CD8⁺ T cells are reflective of CD39⁺ T_{pex} cells. This characteristic aligns with the properties of CD39⁺ T_{pex} cells within blood (Fig. 5d) and the SLOs (Fig. S7B). Collectively, the results support the hypothesis that CD39⁺ T_{pex} cells in the tumor, blood, and lymphoid tissues, characterized by high proliferative activity, are closely associated with ICB-mediated anti-tumor responses and play a critical role in therapeutic efficacy.

page 16, lines 2-3 (subheading)

Intratumoral CD39⁺ T_{pex} cells exhibit consistent high proliferative activity under ICB therapy

page 20, lines 5-10 (discussion section)

Moreover, the IMC analysis post-ICB therapy was constrained by a small number of cases, late time points, and the inclusion of metastatic tissues. Given the limitations of this observational study, direct evidence regarding the changes in the TME triggered by ICB therapy remains lacking. Further verification is needed to understand how CD39⁺ T_{pex} cells are activated following ICB and how they contribute to antitumor activity within the TME.

At several points in the manuscript there is tendency to state that changes in Tex populations are causing clinical responses to therapy (e.g. subheading on p12). The study is observational and cannot make claims of this nature, only that these changes are associated with clinical response: causation cannot be proved from the existing data, so more cautious wording is needed.

Response

Thank you for your valuable comments. As you correctly pointed out, given that our study is observational, we must be cautious with our wording to avoid implying causation where it cannot be proven. We have revised the relevant sections in the manuscript to reflect this. Specifically, we have made changes to the following subheading: page 6, line 14 (affect→are associated with), page 10, lines 18-19 (affect→is associated with), page 13, lines 21-22(affects→is associated with).

We appreciate your careful review and believe these changes will enhance the accuracy and clarity of our manuscript.

page 6, line 14 (subheading)

Tumor CD39+ PD-1+ Tex cells are associated the clinical benefit of ICB therapy

page 10, lines 18-19 (subheading)

Presence of TLSs comprising abundant CD39+ Tpex cells is associated with the hierarchical patterns of CD39+ PD-1+ Tex cells within the TME

page 13, lines 21-22 (subheading)

The selective increase in circulating proliferative CD39+ Tpex cells following ICB therapy is associated with clinical benefit

Minor points:

Figure 1. Some acronyms are undefined in the legend or elsewhere: SLO, NE, NA

Response

Thank you for pointing this out. We have addressed the minor points regarding undefined acronyms. We have defined SLO at page 14, line 12 (subheading), NE at page 22, line 13 (method section) in main manuscript, and added explanations for SLO (page 55, line 2), NE (page 55, line 8), and NA (page 55, line 8) in the Figure 1 legend.

Figure 2. Title: these data do not show that the CD39+ PD-1+ TCF-1+ CD8+ T cells improve clinical outcome of anti-PD-1 therapy, but rather that they are associated with improved outcome.

Response

Thank you for pointing this out. Your comment aligns with the previous feedback we received, and we agree with this perspective. We have made the necessary corrections (page 57, lines 1-3), and we appreciate your careful review.

page 57, lines 1-3 (figure legend)

Fig. 2. CD39⁺ PD-1⁺ CD8⁺ T cells with an exhausted phenotype constitute a TCF-1⁺ exhausted population that is associated with clinical outcomes of anti-PD-1 therapy.

We look forward to hearing from you regarding our submission. We would be glad to respond to any further questions and comments that you may have.

REVIEWERS' COMMENTS

Reviewer #1 (Remarks to the Author):

The revised manuscript meets and exceeds my review comments.

Reviewer #2 (Remarks to the Author):

The authors have addressed most of the raised concerns during this round of revision.

Reviewer #3 (Remarks to the Author):

The authors have revised this paper to satisfactorily address my previous concerns. The paper now makes a valuable contribution to the literature concerning the role of tertiary lymphoid structures in modifying the immune response to tumours in esophageal squamous cell cancer.

Authors Response to Second Round Comments:

Reviewer #1 (Remarks to the Author):

The revised manuscript meets and exceeds my review comments.

Response:

Thank you for taking the time to provide constructive comments on our IMC analysis methods and for offering such valuable suggestions. We sincerely appreciate your guidance, which has greatly improved our study.

Reviewer #2 (Remarks to the Author):

The authors have addressed most of the raised concerns during this round of revision.

Response:

Thank you for taking the time to provide the necessary perspectives and analyses for our study. We deeply appreciate your guidance, which has further strengthened our research by supporting it with additional analysis of the dataset.

Reviewer #3 (Remarks to the Author):

The authors have revised this paper to satisfactorily address my previous concerns. The paper now makes a valuable contribution to the literature concerning the role of tertiary lymphoid structures in modifying the immune response to tumours in esophageal squamous cell cancer.

Response:

Thank you for taking the time to provide constructive comments to improve our study. Your feedback has allowed us to further strengthen the reproducibility and significance of our mass flow cytometry data analysis. We sincerely appreciate your support.